# The transcriptional elongation rate regulates alternative polyadenylation in yeast

**Joseph V Geisberg[†], Zarmik Moqtaderi[†], Kevin Struhl***

Department of Biological Chemistry and Molecular Pharmacology, Harvard Medical School, Boston, United States

**Abstract** Yeast cells undergoing the diauxic response show a striking upstream shift in poly(A) site utilization, with increased use of ORF-proximal poly(A) sites resulting in shorter 3' mRNA isoforms for most genes. This altered poly(A) pattern is extremely similar to that observed in cells containing Pol II derivatives with slow elongation rates. Conversely, cells containing derivatives with fast elongation rates show a subtle downstream shift in poly(A) sites. Polyadenylation patterns of many genes are sensitive to both fast and slow elongation rates, and a global shift of poly(A) utilization is strongly linked to increased purine content of sequences flanking poly(A) sites. Pol II processivity is impaired in diauxic cells, but strains with reduced processivity and normal Pol II elongation rates have normal polyadenylation profiles. Thus, Pol II elongation speed is important for poly(A) site selection and for regulating poly(A) patterns in response to environmental conditions.

***For correspondence:**
kevin@hms.harvard.edu

[†]These authors contributed
equally to this work

**Competing interest:** See
page 20

**Reviewing editor:** Eric J
Wagner, University of Texas
Medical Branch at Galveston,
United States

## Introduction

In eukaryotes, transcription by RNA polymerase II (Pol II) and subsequent RNA processing steps give rise to numerous same-gene mRNA isoforms. These isoforms can exhibit substantial differences in sequence due to alternative splicing, differential 5' and/or 3' end utilization, and other co- and post-transcriptional processes (*Geisberg et al., 2014*; *Berkovits and Mayr, 2015*; *Floor and Doudna, 2016*). This broad mRNA isoform repertoire is important for proper cellular regulation of protein isoform composition, synthesis rate, and localization (*Mayr, 2016*).

Alternative cleavage/polyadenylation in 3' untranslated regions (3'UTRs) is an important mechanism for generating same-gene mRNA isoforms. Most eukaryotic mRNAs are cleaved and polyadenylated at multiple locations within 3'UTRs to generate same-gene isoforms that can be separated by as little as a single nt or by as much as several kb (*Ozsolak et al., 2010*; *Sherstnev et al., 2012*; *Moqtaderi et al., 2013*; *Pelechano et al., 2013*). In *S. cerevisiae* and related yeast species, a typical gene possesses ~60 mRNA 3' isoforms, the vast majority of which are found within the first ~300 nt of the 3'UTR (*Moqtaderi et al., 2013*).

3'UTR regions contain binding sites for proteins and (in many eukaryotes) microRNAs that affect the function of the bound mRNAs (*Bartel, 2009*; *Baltz et al., 2012*; *Freeberg et al., 2013*). Thus, same-gene isoforms that contain or lack particular 3'UTR sequences can differ in their protein and microRNA binding sites, leading to differences in translation efficiency, intracellular localization, and mRNA stability (*Mayr, 2016*). In yeast, same-gene 3' mRNA isoforms, even those that differ by 1–3 nt, can possess different half-lives, in vivo structures (based on DMS profiling), and poly(A)-binding protein (Pab1) binding levels (*Moqtaderi et al., 2018*). Sequences responsible for isoform-specific structures, differential Pab1 binding, and mRNA stability are evolutionarily conserved, indicating biological function (*Moqtaderi et al., 2018*).

Alternative polyadenylation is regulated on a transcriptome scale by environmental or developmental conditions. For example, cancer cells and pluripotent stem cells preferentially express shorter 3′ mRNA isoforms, whereas differentiated cells preferentially express longer 3′ mRNA isoforms (*Mayr and Bartel, 2009*; *Weill et al., 2012*; *Elkon et al., 2013*; *Li and Lu, 2013*; *Tian and Manley, 2013*; *Masamha et al., 2014*). The mechanism by which thousands of genes undergo regulated polyadenylation is poorly understood but is thought to involve mis-regulation of cleavage/polyadenylation factors. Upregulation of CSTF64, FIP1, NUDT21, and PCF11 favors more proximal isoforms, while numerous factors, including CFI25/NUDT21, CPSF6, CFI68, and ELAVL2/3, enhance the usage of more distal poly(A) sites (*Ogorodnikov et al., 2018*; *Kamieniarz-Gdula et al., 2019*). In yeast, alternative polyadenylation in response to environmental conditions has been observed (*Sparks et al., 1997*; *Sparks and Dieckmann, 1998*; *Graber et al., 2013*). Pcf11 and the CPF complex are important contributors to cleavage/polyadenylation site selection, as their inactivation or down-regulation causes a downstream shift in poly(A) sites (*Graber et al., 2013*; *Liu et al., 2017b*; *Gruber and Zavolan, 2019*). In addition, numerous other RNA binding proteins can affect the distribution of 3′ mRNA isoforms in vivo in a more limited fashion (*Tian and Manley, 2017*).

Transcriptional elongation is mechanistically linked to post-transcriptional processes such as splicing, polyadenylation, nuclear export, and histone modification (*Lei et al., 2001*; *Strässer et al., 2002*; *Krogan et al., 2003*; *Ng et al., 2003*; *Bentley, 2014*; *Wallace and Beggs, 2017*). In yeast cells, Pol II derivatives with slow elongation rates have defects in processivity, the ability of Pol II to travel completely down the gene (*Mason and Struhl, 2005*). Similarly, mammalian cells harboring a Pol II derivative with a slow elongation rate exhibit reduced Pol II density in 3′ UTRs, whereas cells with a fast Pol II mutant show increased Pol II at more distal sequences (*Fong et al., 2015*). In *Drosophila*, a slow Pol II mutant affects poly(A) site selection in 3–5% of genes, with an equal number showing increased upstream or downstream utilization of poly(A) sites (*Liu et al., 2017a*). Pol II elongation rates of individual mammalian genes can vary by more than an order of magnitude and are conserved across cell types (*Veloso et al., 2014*). Kinetic competition between elongating Pol II and the Xrn2 exonuclease, which degrades mRNA after cleavage in the 3′ UTR, affects transcriptional termination a few hundred nucleotides downstream (*Kim et al., 2004*; *Fong et al., 2015*; *Baejen et al., 2017*). Similar competition between elongating Pol II and the cleavage/polyadenylation machinery might also affect the choice of poly(A) sites.

Here we show that cells undergoing the diauxic shift, a metabolic shift preceding stationary phase, exhibit a transcriptome-wide 3′ upstream shift in poly(A) site use, leading to shorter 3′ mRNA isoforms. This upstream shift is strikingly mimicked in strains harboring Pol II derivatives with reduced elongation rates. Conversely, albeit to a lesser extent, strains having Pol II derivatives with increased elongation rates show a downstream shift in poly(A) sites. Like yeast Pol II derivatives with slow elongation rates under standard growth conditions (*Mason and Struhl, 2005*), wild-type Pol II shows a processivity defect in diauxic conditions; this defect is strongly correlated with the magnitude of upstream shift. In contrast, mutant strains defective in Pol II processivity but with normal elongation rates display normal patterns of polyadenylation. Thus, Pol II speed influences alternative polyadenylation, and it likely explains the poly(A) pattern changes that occur in yeast cells undergoing the diauxic shift. We suggest that regulation of the Pol II elongation rate in response to environmental or developmental changes represents a novel mechanism to reprogram the 3′ mRNA isoform repertoire that is distinct from changes in the cleavage/polyadenylation machinery.

## Results

### Poly(A) sites are shifted upstream under diauxic conditions, favoring shorter 3′ isoforms

To examine mRNA 3′ isoform distribution as a function of growth condition, we grew duplicate *S. cerevisiae* cultures to mid-log phase in glucose-containing rich medium (YPD), galactose-containing rich medium (YPGal), nutrient-poor minimal medium (MM), and YPD containing 1M sorbitol (Sorbitol), an inducer of osmotic stress. In addition, we examined the early stages of diauxic shift, a condition in which the primary carbon source (glucose) and other metabolites are depleted, by growing the cells for 3 days in YPD (*Vivier et al., 1997*; *Galdieri et al., 2010*). We used 3′ READS to map 3′ mRNA isoforms at single nucleotide resolution on a transcriptome scale (~20 million reads/sample).

Biological replicates exhibit very high correlation to one another on a combined gene expression basis (R = 0.91 to > 0.99; *Figure 1—figure supplement 1A*) as well as at the individual isoform level (R = 0.88 to > 0.99; *Figure 1—figure supplement 1B,C*).

As expected, the total number of sequence reads within a given 3' UTR can vary under different conditions, reflecting regulated expression of many genes under one or more conditions. However, this work focuses on poly(A) profiles, not overall expression levels, of individual yeast genes. Thus, for each gene, we define the level of the most highly expressed 3' mRNA isoform to be 100 and use this to normalize the levels of all other isoforms of the same gene.

The span of genomic sequence encompassing all of a given gene's poly(A) sites is termed the gene's 'end zone,' and a yeast end zone may contain upwards of 60 3' isoform endpoints (*Moqtaderi et al., 2013*; *Pelechano et al., 2013*) across conditions (*Figure 1A*). To avoid problems related to sequencing depth, we limited our analyses to genes with >1000 sequencing reads. Within this subset, we generally focused on each gene's major isoforms, which we define as being at least 5% as abundant as that gene's most highly expressed isoform. On average, a yeast gene gives rise to eight major isoforms (representing ~30% of all 3'UTR polyadenylation sites) under a variety of growth conditions. The section of the end zone occurring between the most proximal and most distal major isoform endpoints is termed the 'major end zone', and the length between these boundaries is the 'major end zone span' (*Figure 1A*). For simplicity, we will sometimes summarize the characteristics of an end zone by referring to the major end zone span and boundaries, the maximally expressed isoform endpoint, and the weighted average isoform endpoint (the arithmetic mean of all isoform endpoints in the end zone).

The isoform distributions of individual gene end zone profiles are very similar in YPD, YPGal, MM, and Sorbitol (*Figure 1B*). Meta-gene plots are nearly superimposable across conditions (*Figure 1C* and *Figure 1—figure supplement 1D*), and end zone parameters such as maximally expressed isoform position (max position), weighted average coordinate, major end zone boundary coordinates, and major end zone span are nearly identical (*Figure 1D*). Thus, growth conditions that greatly affect expression levels of numerous genes do not alter the overall patterns of steady-state isoforms.

In contrast, the end zone patterns of genes in diauxic conditions display a very significant shift in the 5' direction (*Figure 1B*). The maximally expressed isoform position, major end zone boundary coordinates and the weighted average poly(A) position are all shifted 10–25 nt upstream (*Figure 1D*), while the average end zone pattern is clearly different from all the other conditions (*Figure 1C* and *Figure 1—figure supplement 1D*). More than 80% of all genes assayed display an obvious 5' end zone shift (see below), indicating that the altered poly(A) isoform pattern seen in diauxic conditions is a general, genome-wide phenomenon that is independent of overall expression level.

## Yeast cells containing Pol II derivatives with slow elongation rates show a poly(A) pattern that strikingly resembles the pattern in diauxic shift

Pol II derivatives with slow elongation rates have reduced processivity, such that some Pol II molecules dissociate prematurely from the DNA template (*Mason and Struhl, 2005*; *Fong et al., 2015*). As elongating Pol II complexes are exceptionally stable and hence unlikely to simply dissociate, premature Pol II dissociation is likely to be an active process such as cleavage/polyadenylation followed by exonucleolytic RNA degradation, as described in the torpedo model (*Kim et al., 2004*; *Baejen et al., 2017*). In this view, a slow Pol II would give extra time for the cleavage/polyadenylation machinery to function, leading to a 5' bias in poly(A) sites. By analogy, Pol II derivatives with a slow elongation rate show a 5' bias in alternative splicing (*de la Mata et al., 2003*; *Dujardin et al., 2014*). We therefore considered whether the upstream shift of poly(A) sites under diauxic conditions might be a consequence of reduced Pol II processivity and/or slower speed.

To examine the influence of Pol II elongation rate on isoform distribution, we generated yeast strains with mutations (either H1085Q or F1086S) in the largest Pol II subunit (Rpb1) that decrease the elongation rate (*Braberg et al., 2013*). These mutations lie within the 'trigger loop' region of Rpb1, a region that physically interacts with the non-templated strand and is important for ribonucleotide selection, catalysis, and Pol translocation II along the template strand (*Wang et al., 2006*; *Kaplan, 2013*; *Barnes et al., 2015*). We refer to these alleles by their relative speeds with respect to wild-type Pol II: 'slow' (F1086S; 2.5-fold slower than wild-type) or 'slower' (H1085Q; 5-fold slower than wild-type) (*Kaplan et al., 2012*). mRNA 3' isoform profiling of these strains reveals a substantial

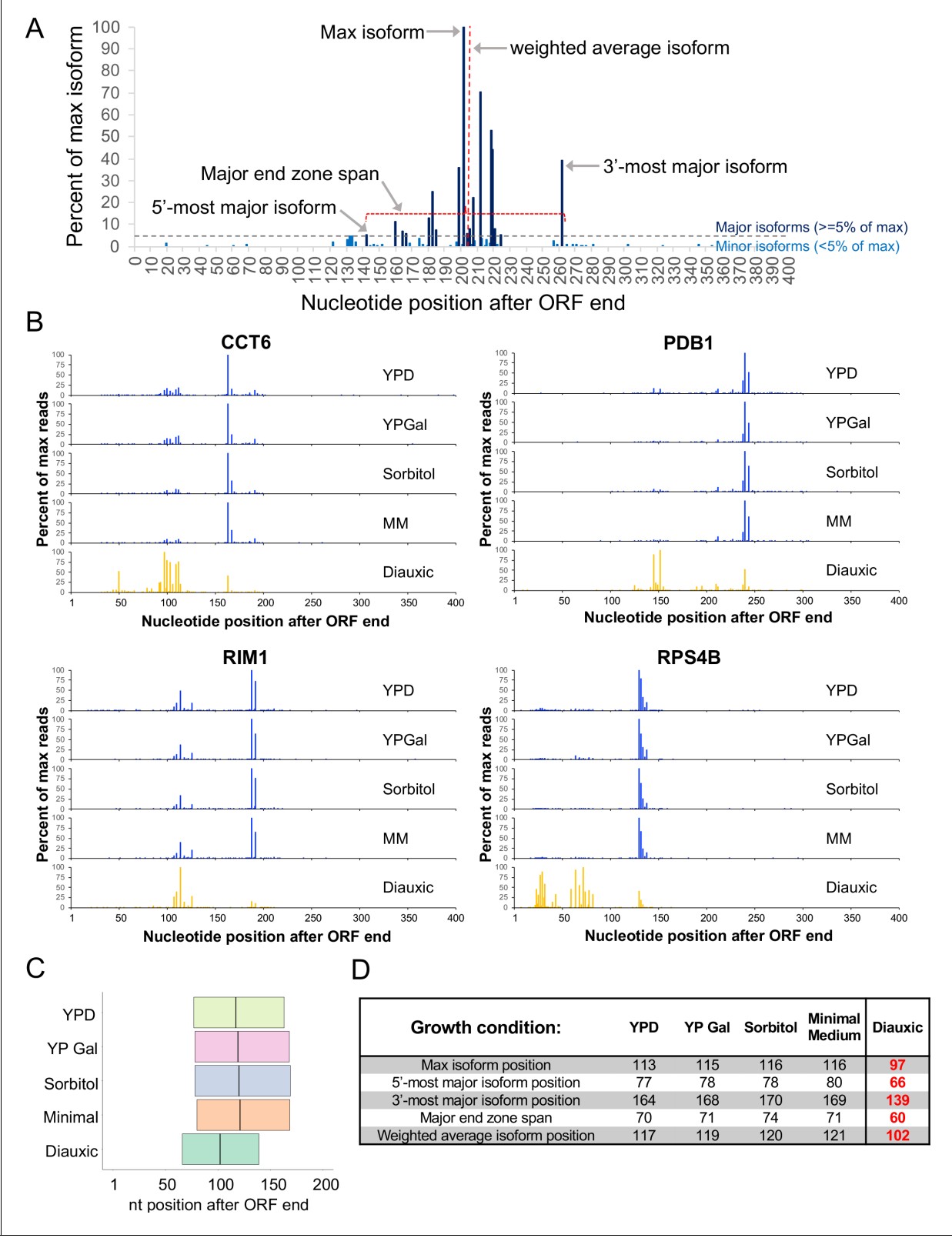

**Figure 1.** Poly(A) sites are shifted upstream in diauxic cells. (**A**) Representative end zone profile (histogram of isoform frequencies) with key landmarks indicated. (**B**) End zone profiles for four genes under five growth conditions. (**C**) Major end zone under five growth conditions. Boundaries represent median values genome-wide for 5'-most and 3'-most major isoforms, and the vertical line within the major end zone represents the genome-wide median of the weighted average isoform position. (**D**) Table of statistics for landmark positions under five growth conditions. Numbers are the median

*Figure 1 continued on next page*

*Figure 1 continued*

values across genes with a combined read count of at least 1000 in both replicates in every condition. Numbers in bold red are shifted upstream from WT in a statistically meaningful way (p < 0.01).

The online version of this article includes the following figure supplement(s) for figure 1:

**Figure supplement 1.** Correlation of biological replicates.

upstream end zone shift that is nearly as dramatic as that observed in diauxic conditions (*Figure 2*). Globally, most end zone parameters are shifted ~10–20 nt upstream relative to those of the wild-type isogenic strain, and the averaged meta-gene end zone profile most closely resembles that of the wild-type strain in diauxic conditions (*Figure 2B and C*, and *Figure 2—figure supplement 1*). Furthermore, 90% of individual genes with an upstream shift in diauxic conditions also show upstream shifts in both slow-Pol II strains (*Figure 2C*), an outcome that is extremely unlikely to occur by chance (p<$10^{-100}$). These results suggest a mechanistic relationship between diauxic conditions and slow Pol II elongation in establishing the pattern of polyadenylation.

The upstream shift in the slow Pol II strains and in diauxic conditions reflects altered poly(A) site utlization per se, because the formal possibility that it is due to preferential degradation of longer mRNA isoforms is highly unlikely. mRNA stability in yeast involves many hundreds of stabilizing and destabilizing elements that are located anywhere within 3'UTRs (*Geisberg et al., 2014*; *Gupta et al., 2014*). As such, longer isoforms within a gene can be either more or less stable than shorter isoforms. Furthermore, the same poly(A) sites are used in normal and diauxic conditions (*Figures 1B* and *2A*, and see below), and it is extremely unlikely that Pol II speed affects the stability of an mRNA isoform, because Pol II must proceed at least 10 nt beyond the poly(A) site in order for this site within the mRNA to become accessible to the cleavage/polyadenylation machinery.

## Upstream shifts involve differential utilization of pre-existing poly(A) sites

We examined whether the upstream-shifted isoforms that predominate in diauxic conditions occur at new polyadenylation positions or represent increased utilization of ORF-proximal sites observed in other conditions. We developed a mathematical model to quantify the likelihood that any overlap in poly(A) positions between exponential growth and diauxic or slow-polymerase conditions is due to chance. Assuming that cleavage/polyadenylation can occur at any position within the first 400 nt of the 3' UTR, the probability that the observed positional overlap occurs by chance is infinitesimal (median R = $1.42 \times 10^{-10}$) (*Figure 3—figure supplement 1*). If we are more conservative and assume instead that the universe of possible cleavage/polyadenylation sites is restricted to sites actually observed in at least one of our growth conditions, then the probability of positional overlap by chance is still vanishingly small (median R = $7.63 \times 10^{-7}$; *Figure 3* and *Figure 3—figure supplement 1*). Thus, the end zone shift generally represents a rebalancing of poly(A) site use rather than the creation of new sites. In accord with these results, the nucleotide frequencies surrounding the major end zone boundary positions and the maximally expressed isoform endpoint are nearly identical under diauxic and slow Pol II conditions (see below). The striking similarity of major isoform positions and nucleotide compositions across conditions indicates that local poly(A) site specificity is mechanistically defined and determined by the basic properties of Pol II and the cleavage/polyadenylation machinery.

## Pol II derivatives with fast elongation rates show modest downstream shifts in poly(A) patterns

As a complement to the above experiments, we also performed mRNA isoform profiling in strains harboring Rpb1 derivatives (L1101S 'fast,' or E1103G, 'faster') whose Pol II elongation rates are 2- to 2.5-fold faster than wild-type (*Kaplan et al., 2012*; *Braberg et al., 2013*). L1101 and E1103 lie within an α-helical region adjoining the trigger loop that is thought to contact the non-templated DNA strand and be important for Pol II translocation. In comparison to the slow Pol II strains, these fast Pol II strains show more subtle changes to 3' isoform distributions, with small but significant downstream shifts in end zone parameters such as the maximal position, major end zone span, and weighted average coordinate (*Figure 4*). The Rpb1-E1103G strain exhibits a slightly greater overall

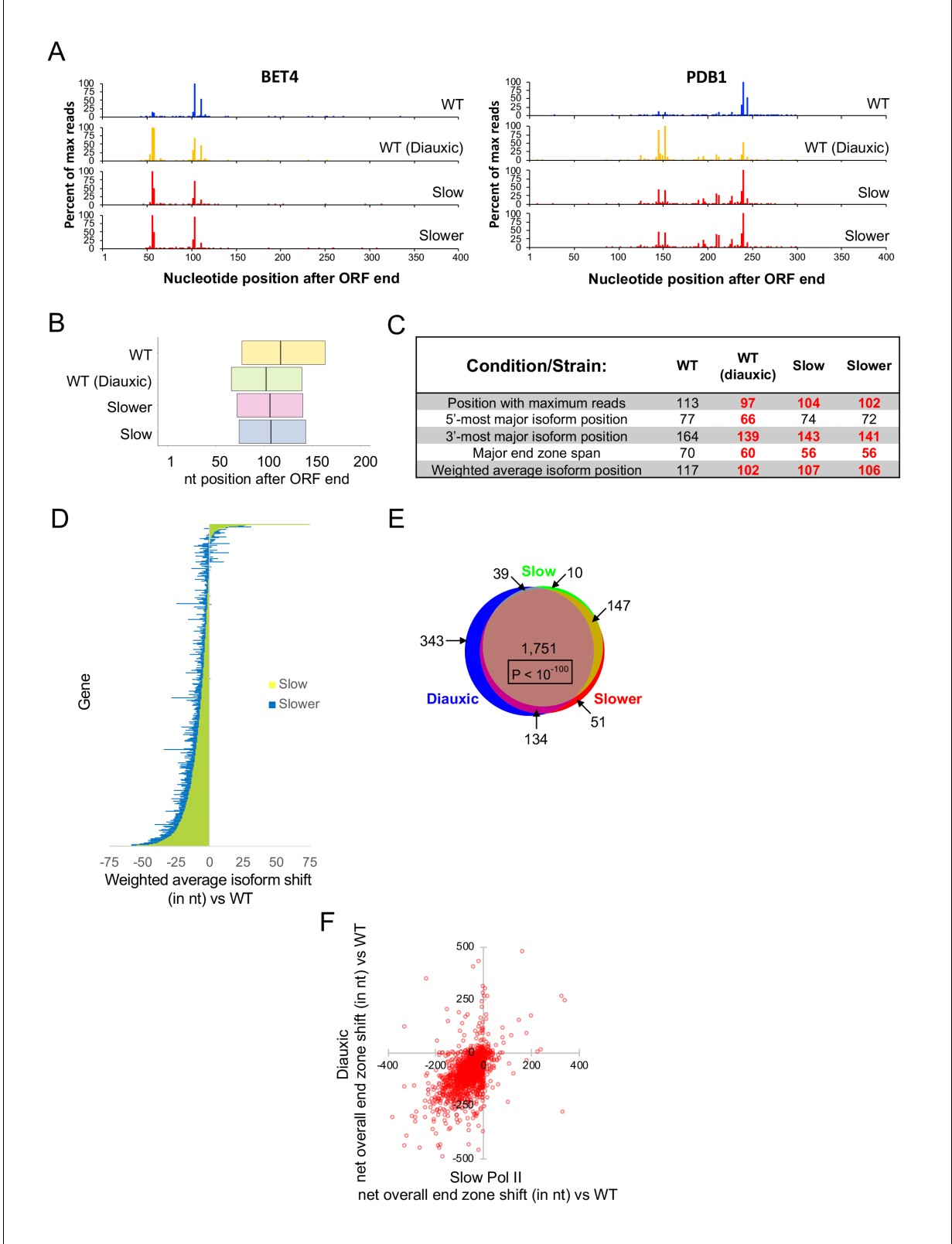

**Figure 2.** Slow Pol II and diauxic end zones are highly similar. (**A**) End zone profiles for *PDB1* and *BET4* in strains harboring wild-type Rpb1 (in exponential and diauxic growth conditions), Rbp1 H1085Q ('slower'), and Rbp1 F1086S ('slow'). (**B**) Major end zones of these strains. Boundaries represent median values genome-wide for 5'-most and 3'-most major isoforms, and the vertical line within the major end zone represents the genome-wide median of the weighted average isoform position. (**C**) Table of statistics for landmark positions. Numbers are the median values across genes with

*Figure 2 continued on next page*

*Figure 2 continued*

a total of at least 1000 sequence reads in both replicates in every condition. Bold red numbers are shifted upstream vs WT in a statistically meaningful way (p < 0.01). (**D**) Bar graph representation of each gene's net shift in weighted average isoform position in strains with slow vs wild-type Rpb1. Each horizontal line represents one gene, ordered by shift values in the 'slow' strain; the graph includes 3497 genes with a combined read count of at least 1000 for both replicates in all three strains. Yellow bars represent the 'slow' strain, and blue is for the 'slower' strain; overlapping bars appear green. To obtain net shift values for every gene in each mutant strain, the average shift vs WT in two replicates was diminished by the absolute value of the average shift of the WT and mutant biological replicates. The net shift was set to zero if the absolute value of the shift vs WT was less than the absolute value of the shift between biological replicates. (**E**) Venn diagram overlap of genes categorized as upshifted in the diauxic condition, slower Rpb1 (H1085Q), or slow Rpb1 (F1086S) strains. (**F**) Correlation of end zone shifts in diauxic and slow Pol II strains. The average net overall end zone shift in slow Pol II strains (x-axis; see Materials and methods) is plotted against the net overall end zone shift in diauxic cells (y-axis). Negative values represent upstream shifts, and positive values indicate downstream end zone shifts.

The online version of this article includes the following figure supplement(s) for figure 2:

**Figure supplement 1.** Heat map of percent coordinate utilization in 3'UTRs.

downstream shift than the Rpb1-L1101S strain, consistent with its faster elongation rate (*Kaplan et al., 2012*; *Braberg et al., 2013*) Similarly, the upstream shifts of the slow Pol II strains parallel their elongation rates determined in vitro, with the slower mutant shifted farther upstream. The relationship of the elongation rates determined in vitro to the poly(A) pattern shift in vivo strongly suggests that the effects on polyadenylation are due to the elongation rate.

## Poly(A) patterns of individual genes vary in their sensitivity to Pol II elongation rate

To address the relationship between poly(A) patterns in the slow and fast Pol II derivatives, we constructed a mathematical error model to determine whether the poly(A) pattern of an individual gene is significantly shifted in either the upstream or downstream direction. In the slow Pol II mutants, 2083 (Rpb1-H1085Q 'slower') and 1947 (Rpb1-F1086S 'slow') genes (out of a total of 2,790) show significant upstream shifts (*Figure 4—figure supplement 1*), with more than 97% of the upstream shifts occurring in both strains (1898 out of 1947 genes, $p<10^{-100}$, hypergeometric test). In the fast Pol II strains, a smaller proportion of genes (23% for Rpb1-L1101S 'fast' and 32% for Rpb1-E1103G 'faster') show downstream shifts, with the vast majority of these shifts (95%; $p<10^{-100}$) occurring in both strains (*Figure 4—figure supplement 1*). Interestingly, 76% (462 out of 605) of genes showing downstream shifts in both fast Pol II strains also exhibit upstream shifts in both slow Pol II strains, an overlap that is highly significant (*Figure 4E and F*, and *Figure 4—figure supplement 1*; $p=2.68\times10^{-7}$). Thus, 17% of yeast genes tested are especially sensitive to perturbations in Pol II elongation rate, both fast and slow. The striking similarities in polyadenylation profiles between the two slow and the two fast Pol II derivatives indicate that these patterns depend on Pol II elongation rate and not on other properties of the Pol II derivatives.

Unexpectedly, a minority class of genes behave in the opposite manner. In the slower Pol II strain, a small number of genes (46; *Figures 2D*, *4E and F*, and *Figure 4—figure supplement 1*) exhibit an atypical downstream shift, 34 of which also show a downstream shift in strains containing the slow Pol II derivative ($p=6.0\times10^{-52}$, hypergeometric test). Conversely, in the fast Pol II strains, a small minority of genes show atypical upstream shifts in (196 for Rpb1-L1101S and 228 for Rpb1-E1103G), with ~72% of these showing upstream shifts in both strains ($p<10^{-100}$) (*Figure 4D and E* and *Figure 4—figure supplement 1*). The fact that these opposite patterns are observed in two different strains with the same catalytic properties (either fast or slow Pol II) suggest that a minority of genes have Pol II elongation properties in vivo that are different from one would expect from the Pol II elongation rate determined in vitro on a specific DNA template.

## Sequences around cleavage sites of Pol II speed-sensitive genes are enriched for purines

Although overall nucleotide frequencies at sequences located ±10 nt from poly(A) sites are virtually identical (*Figure 5—figure supplement 1A*), we examined whether such sequences surrounding max isoform endpoints differ between Pol II speed-sensitive genes ('Both' category) and genes unaffected by Pol II elongation rate. Interestingly, speed-sensitive genes have reduced frequencies in U and (to a lesser extent) C residues, and a greater incidence of A and G residues relative to speed-

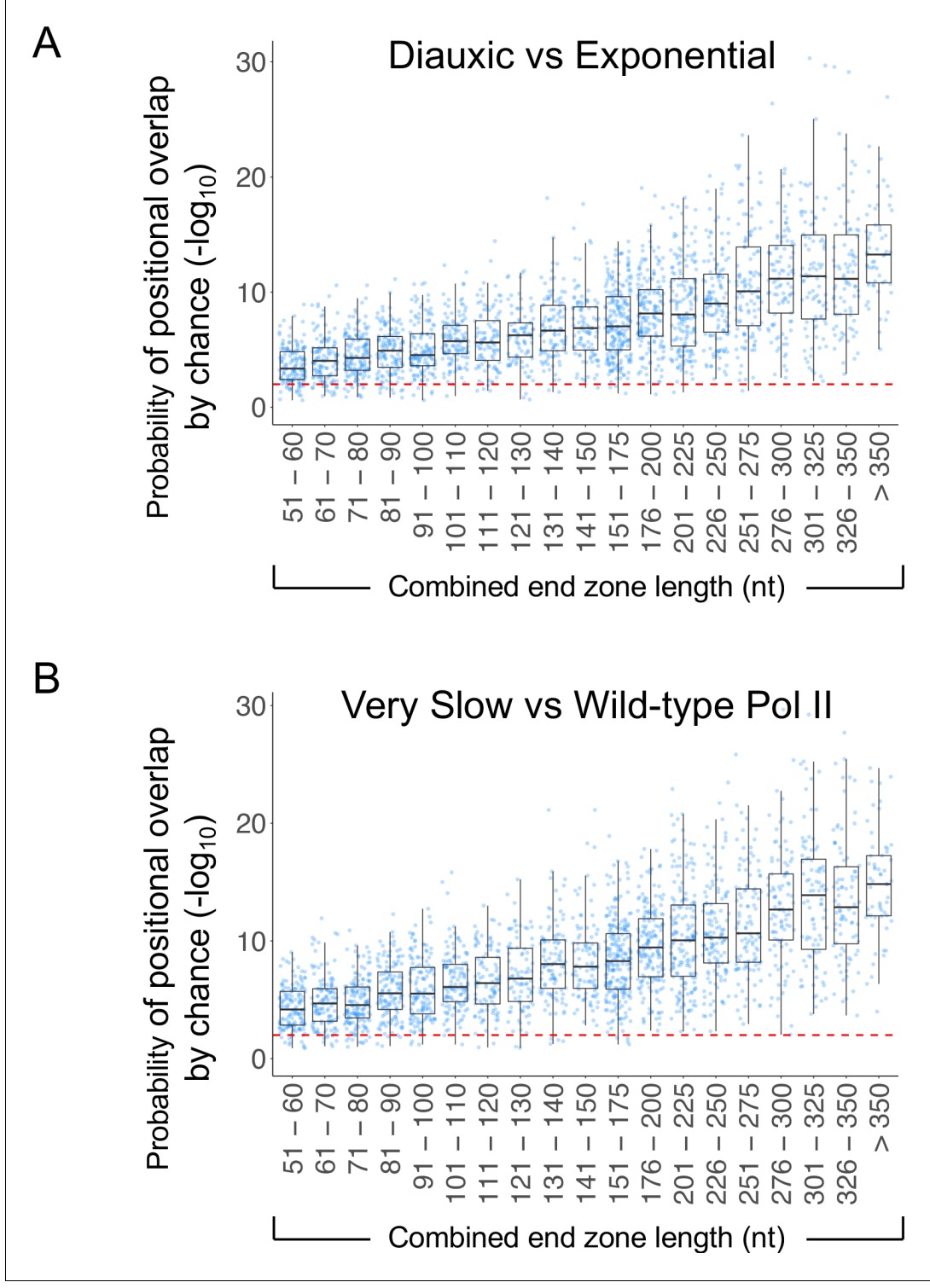

**Figure 3.** High overlap in poly(A) sites used in diauxic and slow-Pol II strains. (**A**) Probability of overlap in isoform distribution by chance as a function of combined end zone length in strains with very slow (H1085Q) or wild-type Rpb1. (**B**) Probability of overlap in isoform distribution by chance as a function of combined end zone length in exponential growth and diauxic conditions.

The online version of this article includes the following figure supplement(s) for figure 3:

**Figure supplement 1.** High poly(A) site overlap across strains/conditions despite differences in relative levels.

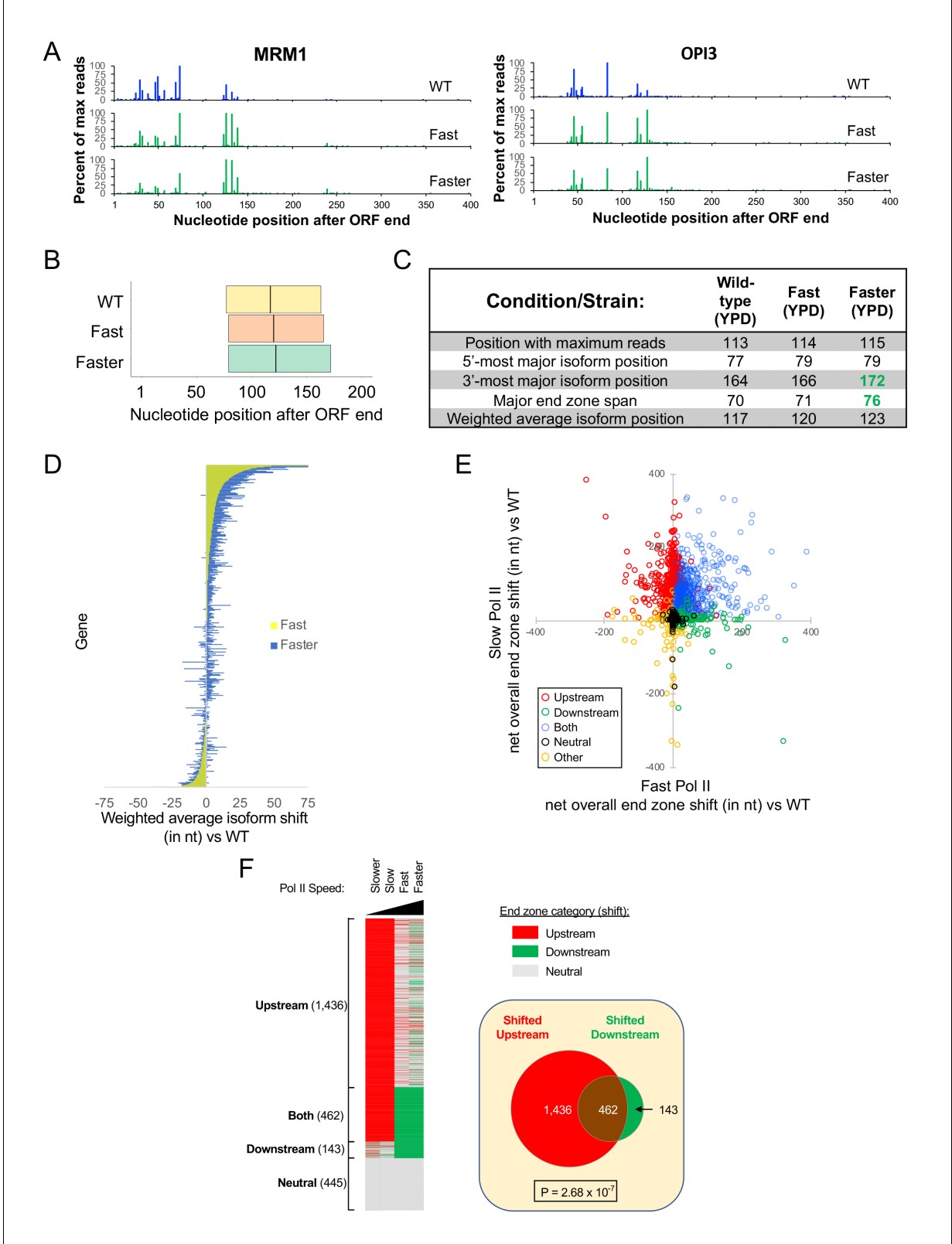

**Figure 4.** Increased usage of downstream poly(A) sites in fast Pol II strains. (**A**) End zone profiles for *MRM1* and *OPI3* in strains with wild-type, L1101S ('fast'), and E1103G ('faster') Rpb1. (**B**) Major end zones of these strains. Boundaries represent median values genome-wide for 5'-most and 3'-most major isoforms, and the vertical line within the major end zone represents the genome-wide median of the weighted average isoform position. (**C**) Table of statistics for landmark positions. Numbers aremedian values across genes with a total of at least 1000 sequence reads in both replicates in

*Figure 4 continued on next page*

*Figure 4 continued*

every condition. Numbers in bold green are significantly shifted downstream from WT (p < 0.01). (D) Bar graph representation of each gene's net shift in weighted average isoform position in strains with fast vs wild-type Rpb1. Each horizontal line represents one gene, ordered by shift values in the 'fast' strain; the graph includes 3627 genes with a combined read count of at least 1000 for both replicates in all three strains. Yellow represents the 'fast' strain and blue the 'faster' strain, with the overlap appearing green. To obtain net shift values for every gene in each mutant strain, the average shift vs WT in two replicates was diminished by the absolute value of the average shift of the WT and mutant biological replicates. The net shift was set to zero if the absolute value of the shift vs WT was less than the absolute value of the shift between biological replicates. (E) 2790 genes are plotted as a function of the average overall net end zone shift (see Materials and methods) in either catalytically fast (x-axis) or slow (y-axis) Pol II mutants. Genes were classified into Upstream (red), Downstream (green), Both (blue), Neutral (black) and Other (orange) on the basis of each gene's net end zone shift (see text). The upper right-hand quadrant comprises genes shifted upstream in slow Pol II mutants and downstream in fast Pol II mutants, while genes in the upper left-hand quadrant are shifted upstream in both fast and slow Pol II mutant strains. The bottom right quadrant contains genes that are shifted downstream in both slow and fast Pol II mutants, while the few genes whose end zones are shifted downstream in slow Pol II and upstream in fast Pol II strains are found in the bottom left quadrant. (F) Left: Classification of genes by category. The categories are: 'Upstream,' genes whose poly (A) sites were upshifted in both slow-Pol II strains; 'Downstream,' genes whose end zone profiles were downshifted in both fast-Pol II strains; 'Neutral,' genes with no end zone shift in any slow or fast Pol II-containing strain; and 'Other,' genes with any other combination of properties (see Materials and methods). Right: Venn diagram illustrating the 'Both' sub-category of genes (see Materials and Methods), i.e. the intersection of the set of genes shifted upstream in slow Pol II (Upstream category) with the set of genes shifted downstream in the presence of fast Pol II (Downstream category).

The online version of this article includes the following figure supplement(s) for figure 4:

**Figure supplement 1.** Downstream end zone shift in fast Pol II strains.

unaffected genes (*Figure 5*). This skewed frequency of purines to pyrimidines is observed in all conditions tested, and nucleotide distributions in speed-sensitive genes bear a striking semblance to one another irrespective of condition or speed category (*Figure 5—figure supplement 1C*). This is noteworthy because max isoform positions (and hence adjacent sequences) vary greatly among conditions and gene categories (*Figure 5—figure supplement 1B and D*). These results strongly suggest that localized sequence composition, not location within the 3'UTR, is the primary determinant of susceptibility to cleavage/polyadenylation changes in strains with altered Pol II elongation rates and in diauxic conditions.

## Evidence that Pol II elongation rate is decreased in diauxic conditions

The striking similarity of poly(A) profiles in diauxic conditions and in strains with slow-elongating Pol II derivatives suggests that the Pol II elongation rate is slower in diauxic conditions than it is in exponentially growing cells. Because diauxic cells are carbon source starved, it is impossible to directly measure the Pol II elongation rate using a conventional assay that involves rapid glucose shutoff of a long gene (*Mason and Struhl, 2005*). Instead, we used Pol II processivity as a proxy for elongation rate, based on the observation that a slow elongation rate is associated with decreased Pol II processivity and disproportionate accumulation at promoter regions in vivo (*Mason and Struhl, 2005*; *Fong et al., 2017*).

We compared Pol II occupancy at the coding sequences and promoter regions of 14 genes in diauxic versus exponentially growing cells. The resulting promoter:ORF occupancy ratios under each condition were combined to generate a diauxic:exponential processivity ratio (*Figure 6A*). Some genes display diauxic:exponential ratios of ~1, indicating that the Pol II distributions are similar in both conditions. However, most of the genes tested have diauxic:exponential ratios ranging from 2 to 6, indicating disproportionate accumulation of Pol II at promoter regions in diauxic conditions. Importantly, the extent of the upstream shift in poly(A) site selection is strongly correlated (*Figure 6B*; R = 0.63) with the diauxic:exponential processivity ratio, suggesting that Pol II elongates slowly under diauxic conditions.

## Pol II elongation rate, not processivity, is important for polyadenylation patterns

The above analysis of Pol II processivity under diauxic and exponential growth conditions cannot distinguish whether the polyadenylation pattern is due to a reduced elongation rate or to a decrease in Pol II processivity. To address the role of Pol II processivity more specifically, we examined the poly (A) profiles of cells that lack either Spt4 or Hpr1, two proteins that travel with elongating Pol II. Spt4

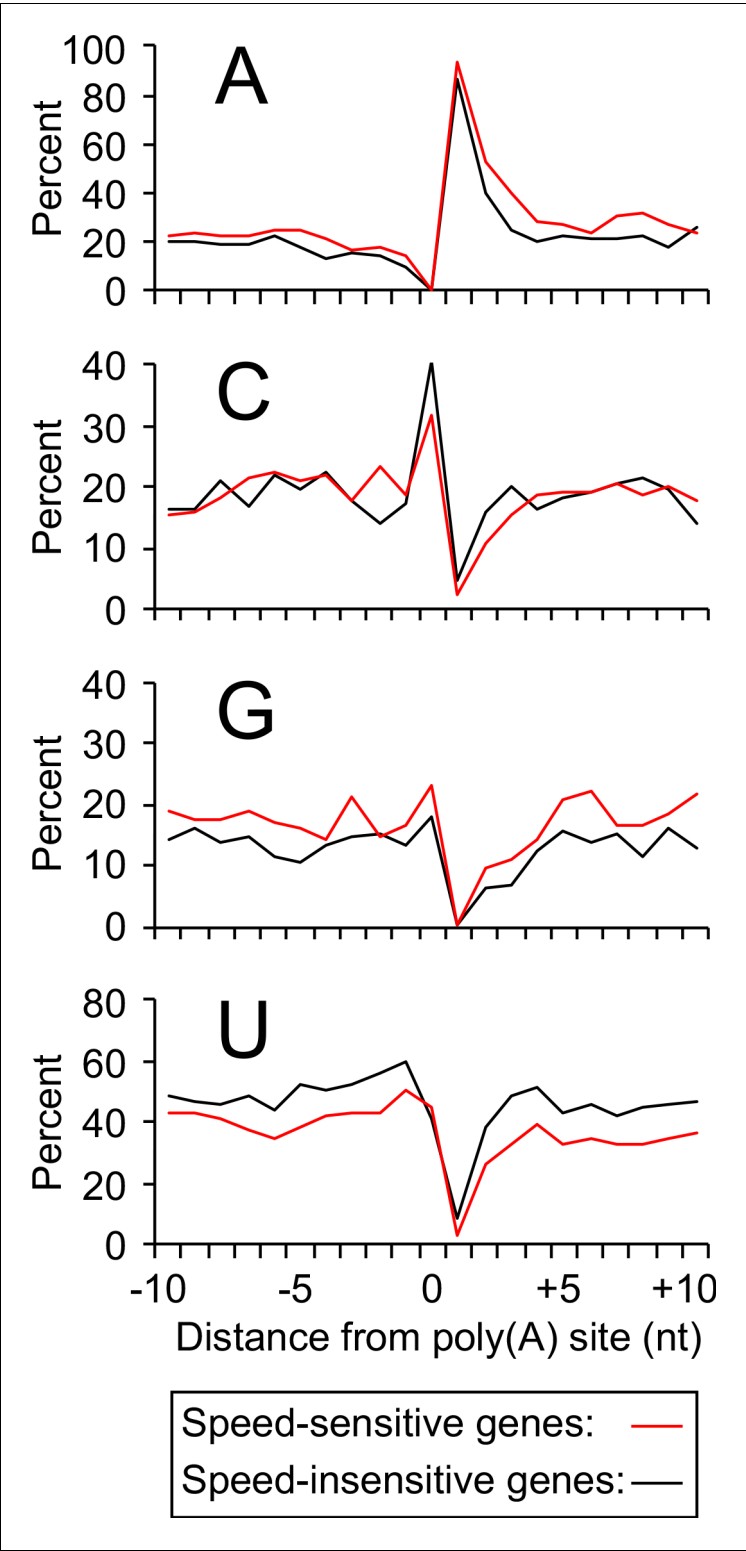

**Figure 5.** Increased purine content in sequences flanking poly(A) sites of genes sensitive to Pol II speed. Nucleotide frequencies near max isoform poly(A) sites in the wild-type strain (exponential culture) for speed-sensitive genes (462 genes belonging to the 'Both' category; red lines) and speed-insensitive genes (445 genes belonging to the 'Neutral' category; black lines).

The online version of this article includes the following figure supplement(s) for figure 5:

**Figure supplement 1.** Percent identity of max isoform positions by condition, strain, and category.

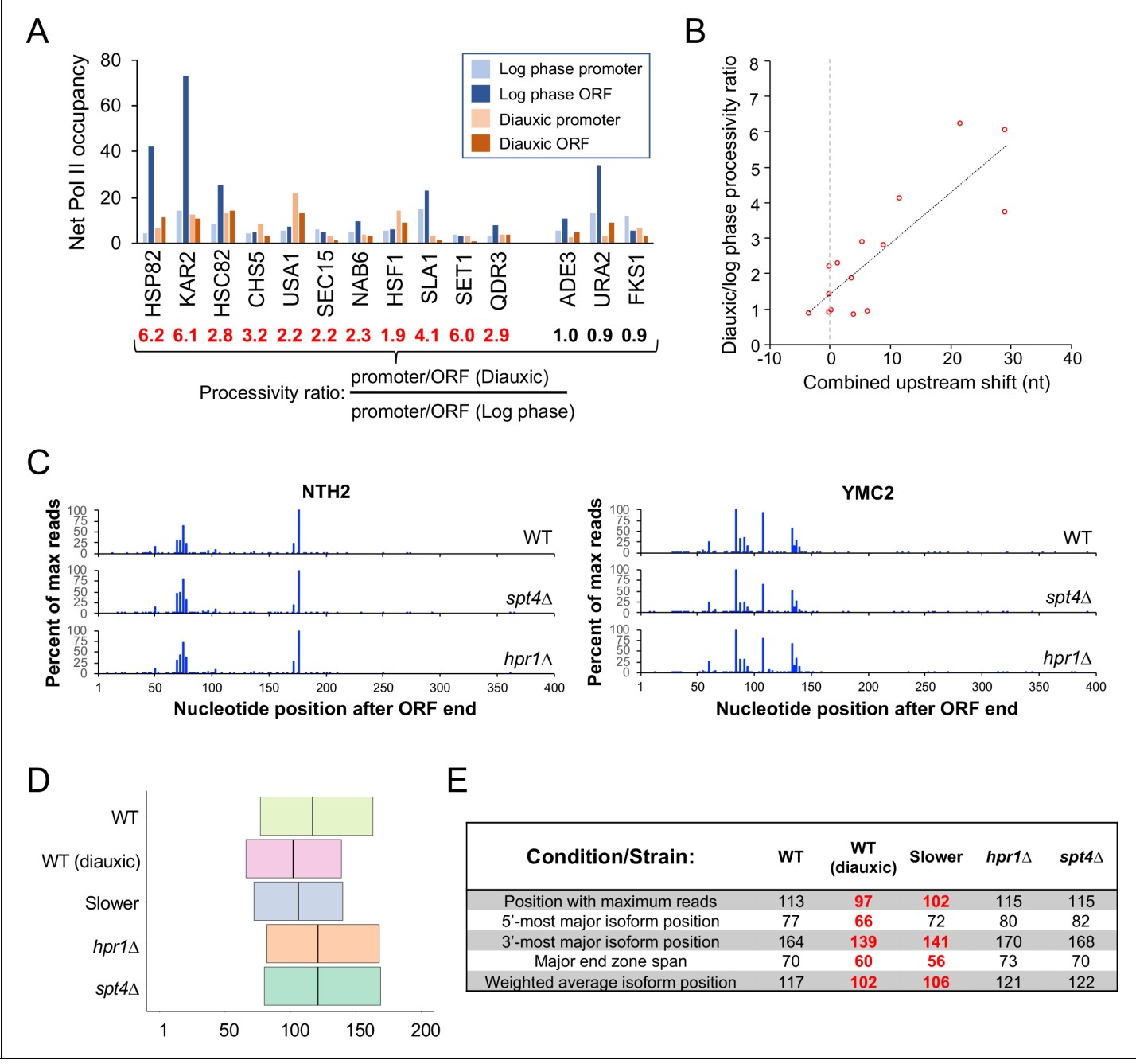

**Figure 6.** Pol II elongation rate is linked to shifted end zone profiles in diauxic conditions. (A) Pol II occupancy (background-subtracted ChIP signal) at promoters and ORFs of select genes in logarithmic growth and diauxic conditions. For every gene, the promoter/ORF occupancy ratio is determined for each condition, and the ratio of these ratios (diauxic/log phase), termed the processivity ratio, is given under the locus name. (B) Scatter plot of the diauxic/log phase processivity ratio vs upstream shift (see Materials and methods) in nt observed in diauxic conditions. (C) End zone profiles of *NTH1* and *YMC2* in wild-type, *spt4Δ*, and *hpr1Δ* strains. (D) Plot of genome-wide median major end zones in wild-type (log phase and diauxic), slower-Pol II (Rpb1 H1085Q), *hpr1Δ*, and *spt4Δ* strains. (E) Landmark statistics table in these strains. (All genes with >1000 reads/condition). Bold red numbers represent statistically meaningful upstream shifts vs WT (p < 0.01).

The online version of this article includes the following figure supplement(s) for figure 6:

**Figure supplement 1.** 3'UTR percent coordinate utilization for several strains/conditions.

and Hpr1 deletion strains exhibit Pol II processivity defects, but they do not affect the Pol II elongation rate (*Mason and Struhl, 2005*). The poly(A) profiles of *spt4Δ* and *hpr1Δ* strains are very similar to those of the wild-type strain at the individual gene level (*Figure 6C*). In fact, meta-gene profiles and various end zone parameters indicate a very modest downstream shift in poly(A) site utilization (*Figure 6D* and *Figure 6—figure supplement 1*). Thus, Pol II processivity per se does not influence poly(A) profiles, arguing that a decrease in the elongation rate is the cause of the upstream shift observed under diauxic growth conditions.

## Discussion

### Pol II elongation rate, not Pol II processivity, affects poly(A) site selection

Pol II elongation is mechanistically linked to post-transcriptional processes such as splicing, polyadenylation, chromatin modification, and mRNA localization. Moreover, yeast and metazoan cells containing Pol II derivatives with slow elongation rates show altered patterns of mRNA splicing and histone modifications throughout the transcribed regions. Here we use multiple Pol II derivatives with slow or fast elongation rates and nucleotide-level analysis to show that the rate of Pol II elongation has a dramatic influence on the pattern of polyadenylation (*Figure 7*).

Two different slow Pol II derivatives cause a near-identical, transcriptome-wide, upstream shift in the relative use of known poly(A) sites, but they do not typically result in multiple novel poly(A) sites. The upstream shifts are observed in the majority of yeast genes, but some genes are unaffected by these Pol II derivatives. In contrast to the slow Pol II derivatives, two fast Pol II derivatives confer a

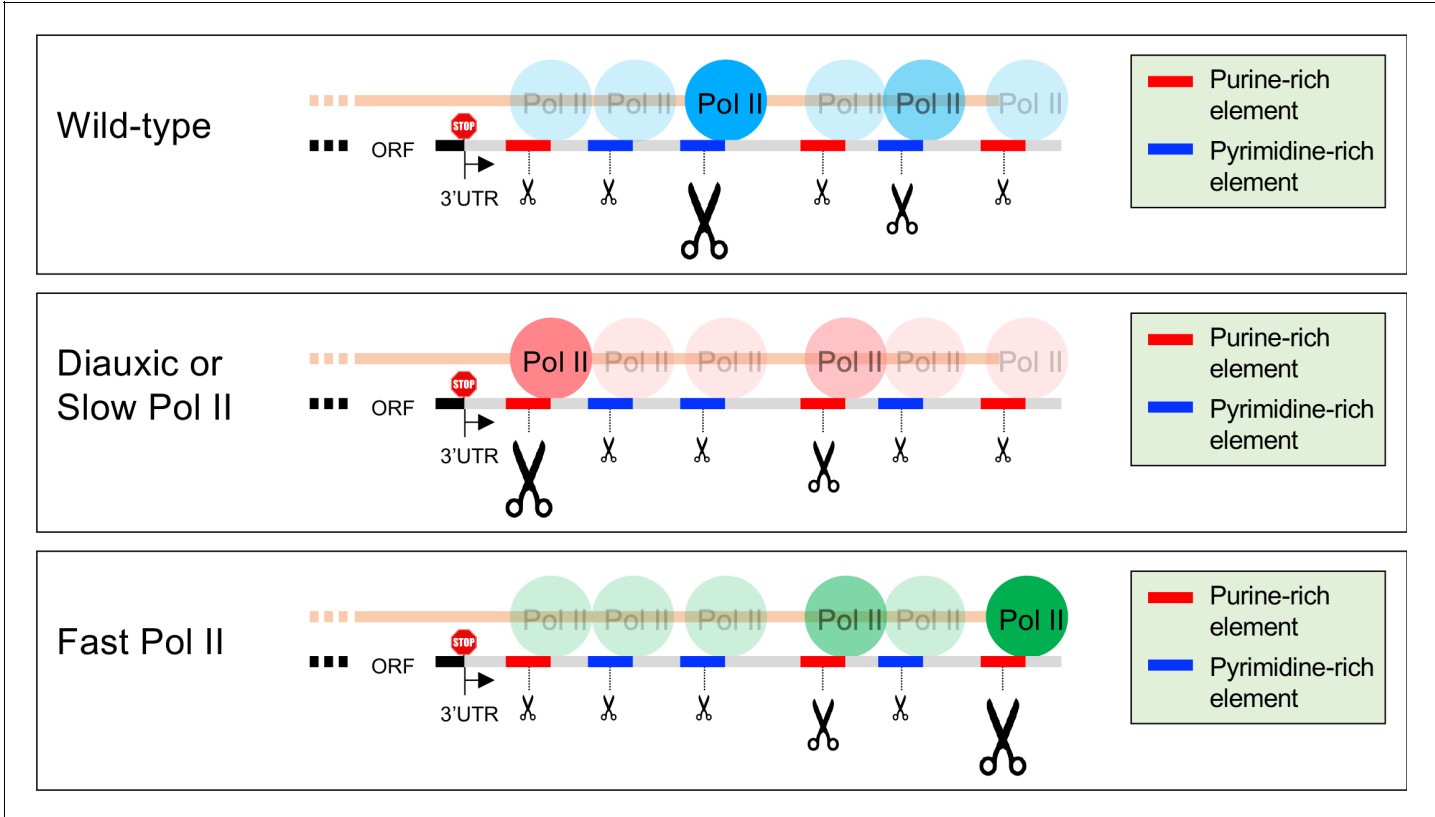

**Figure 7.** Model of poly(A) site shift in Pol II speed-sensitive genes. The 3'UTRs of speed-sensitive genes contain purine-rich elements (red line segments) and pyrimidine-rich elements (blue line segments) of varying strengths (small, medium or large scissors). Under normal conditions (exponentially-growing wild-type cells), cleavage and polyadenylation takes place predominantly at pyrimidine-rich elements. In diauxic conditions and in cells harboring slow Pol II, purine-rich elements drive an upstream shift in polyadenylation patterns, likely due to increased Pol II dwell time at those sequences. Conversely, fast Pol II shifts the poly(A) patterns to more distal purine rich sites.

downstream shift in poly(A) site preference. These downstream shifts seen with the fast Pol II derivatives are much subtler than the upstream shifts observed with the slow Pol II derivatives, in both the number of genes affected and the magnitude of the shifts. Nevertheless, the overlap between upstream- and downstream-shifted genes is far beyond what would be expected by chance, indicating that the poly(A) patterns of many genes are sensitive to both slow and fast Pol II elongation rates. Interestingly, purine-rich sequences flanking cleavage/polyadenylation sites are associated with Pol II genes that are sensitive to fast and/or slow Pol II elongation speed.

Pol II derivatives with slow elongation rates also have defects in Pol II processivity. Strains with *hpr1* or *spt4* deletions exhibit defects in Pol II processivity comparable to those in strains with slow Pol II derivatives, but they have normal elongation rates (*Mason and Struhl, 2005*). In these strains, poly(A) patterns are unaffected, indicating that Pol II elongation rate, not Pol II processivity, is the major determinant of poly(A) patterns in yeast.

## Evidence that regulated polyadenylation during the diauxic shift is due to decreased elongation rate

Yeast cells undergoing the diauxic shift display a transcriptome-wide, upstream-shifted poly(A) pattern that is remarkably similar (though not identical) to the poly(A) patterns conferred by the two Pol II derivatives with slow elongation rates. Although we cannot directly measure the Pol II elongation rate under diauxic conditions, Pol II under these conditions is disproportionately found in promoter regions, a property linked to slow Pol II elongation rate (*Mason and Struhl, 2005*; *Fong et al., 2017*). Importantly, the degree of promoter bias in the Pol II distribution is strongly correlated with the magnitude of the upstream shift. Taken together, our results suggest that upstream-shifted polyadenylation during the diauxic shift is due to a decreased Pol II elongation rate under these conditions.

It is formally possible that the upstream shift in diauxic conditions is due to changes in the biological activity or expression level of a 3' mRNA processing factor. However, this is highly unlikely, because any diauxic-shift-induced alteration in a 3' processing factor would have to result in a poly(A) pattern virtually identical to those of two different slow Pol II mutants over thousands of genes. Furthermore, this explanation does not account for why there is such a pronounced Pol II processivity defect, a hallmark of reduced Pol II speed.

The presumed decrease in Pol II elongation rate under diauxic conditions could be due to the physiological state of the cells and/or modification of Pol II (or an associated elongation factor). Cellular stress or slow growth alone would be unlikely to cause the upstream shift, because other stressful conditions, including those that reduce the growth rate, do not affect the poly(A) pattern (*Figure 1*). However, limitation of a specific nutrient(s) or oxygen could affect the Pol II elongation rate. In this regard, the Pol II elongation rate is reduced in cells treated with 6-azauracil or mycophenolic acid (*Mason and Struhl, 2005*), conditions that reduce intracellular levels of GTP and UTP and hence substrates for transcription.

## Mechanistic implications about regulation of alternative polyadenylation

There are many examples of alternative polyadenylation regulation in response to environmental stress or developmental conditions (*Flavell et al., 2008*; *Sandberg et al., 2008*; *Ji et al., 2009*; *Mayr and Bartel, 2009*). A variety of experiments suggest that such regulation of mRNA 3' end formation involves components of the cleavage/polyadenylation machinery and RNA-binding proteins (*Elkon et al., 2013*; *Tian and Manley, 2017*). This regulation could occur either by altered expression of such components and/or modification that changes their activity. Our work demonstrates that control of the Pol II elongation rate is an alternative mechanism for regulating alternative polyadenylation in response to physiological conditions. Regulation by Pol II elongation rate is not mutually exclusive with regulation of the cleavage/polyadenylation machinery, and indeed both mechanisms could operate under a given physiological condition. It is currently unknown, and hence would be interesting, to examine Pol II elongation rates under situations in which alternative polyadenylation is regulated.

# Materials and methods

## Strains

Mutations in *RPO21* and precise ORF deletions of *HPR1* and *SPT4* were introduced into the JGY2000 strain (**MATa**, *his3Δ0, leu2Δ0, met15Δ0, ura3Δ0, rpb1::RPB1–FRB, rpl13::RPL13–FK512*) (*Geisberg et al., 2014*) by CRISPR, using derivatives of pML104 (*Laughery et al., 2015*) to supply Cas9 and guide RNA. All strains were confirmed by PCR and Sanger sequencing.

| Strain | *RPO21* Allele | Other |
|---|---|---|
| JGY2000 | *RPO21-FRB* | |
| JZY5 | *RPO21-H1085Q-FRB* | |
| JZY6 | *RPO21-F1086S-FRB* | |
| JZY14 | *RPO21-L1101S-FRB* | |
| JZY15 | *RPO21-E1103G-FRB* | |
| JZY27 | *RPO21-FRB* | *spt4Δ* |
| JZY33 | *RPO21-FRB* | *hpr1Δ* |

## RNA analysis

Except for the diauxic condition JGY2000, all strains were grown in 50 ml of media (see below) to $OD_{600}$ = 0.3–0.4 at 30° C. JGY2000, JZY5, JZY6, JZY14, JZY15, JZY27, and JZY33 were grown in YPD. JGY2000 was also grown in YP medium containing 2% Galactose ('YPGal'), osmotic stress conditions (YPD + 1M sorbitol;'Sorbitol') and nutrient poor minimal medium (2% dextrose, yeast nitrogen base with ammonium sulfate and without amino acids supplemented with uracil and essential amino acids; 'MM'). Diauxic conditions were achieved by first growing JGY2000 at 30°C in 50 ml YPD to an $OD_{600}$ = 0.3–0.4 (~24 hr) and then allowing the cells to grow in the same medium for an additional 48 hr (~72 hr total growth time and a final $OD_{600}$ = 3.0). Total RNA was isolated and purified from 15 to 20 ml of cells (10 ml for the diauxic condition) using the hot acid phenol method followed by QIAGEN RNeasy as described (*Moqtaderi et al., 2018*). 3' READS was performed with 25 ug of purified total RNA with 17 cycles of amplification (*Jin et al., 2015*). Barcoded libraries were quantified on an Agilent Bioanalyzer 2100, pooled, and sequenced on the Illumina NextSeq 500 platform.

## Chromatin immunoprecipitation

Whole-cell lysates from 30 ml of formaldehyde-treated cells were prepared as described (*Geisberg et al., 2014*). 150 µl of extracts were diluted to a total volume of 950 µl with FA lysis buffer (*Aparicio et al., 2004*) and immunoprecipitated with 10 µl of 8WG16 antibody (Biolegend #664912) for 2 hr at room temperature. Protein-DNA complexes were then incubated for an additional 2 hr with 50 µl of 50% (v:v in FA lysis buffer) protein A-Sepharose. Beads bound with Pol II-DNA were washed and eluted as described (*Aparicio et al., 2004*). Pol II binding occupancy was assayed by real-time qPCR (*Geisberg et al., 2014*) with oligonucleotides specific to either promoter regions or coding sequences of selected genes (see oligo table below).

| Gene | Location | Position relative to ATG | Sequence |
|---|---|---|---|
| HSP82 | Promoter | −202 | 5'-TGGTTTTATGAGCGGTTAATTC-3' |
| HSP82 | Promoter | −79 | 5'-GGGAAGAAATGAGGAGGTC-3' |
| HSP82 | ORF | 2022 | 5'-GGGTTTGAACATTGATGAGG -3' |
| HSP82 | ORF | 2146 | 5'-GGCCATGATGTTCTACCTAA-3' |
| HSC82 | Promoter | −120 | 5'-GAACTGCCTACCGTAAGTG-3' |
| HSC82 | Promoter | −27 | 5'-GGTTCTGTAGCGTTTCAAGA-3' |

*Continued on next page*

*Continued*

| Gene | Location | Position relative to ATG | Sequence |
|------|----------|--------------------------|----------|
| HSC82 | ORF | 1931 | 5'-AGACCGCTTTGTTGACTTC-3' |
| HSC82 | ORF | 2048 | 5'-GCGGTTTCTGTTTCTTCATC-3' |
| URA2 | Promoter | −177 | 5'-ATAGAGATCTTCATGGCACG-3' |
| URA2 | Promoter | −53 | 5'-AGTTATGGATTTCTATCGTCGT-3' |
| URA2 | ORF | 2029 | 5'-GTAGCCCCATCTCAAACTTT-3' |
| URA2 | ORF | 2124 | 5'-ACATTCACCAACAACACCTA-3' |
| ADE3 | Promoter | −141 | 5'-CATTATATACGCGCTCTCCA-3' |
| ADE3 | Promoter | −20 | 5'-AAGTTGTGTTCGTCTCGTTA-3' |
| ADE3 | ORF | 1951 | 5'-GCCTCTTCTGTTATTGCTGA-3' |
| ADE3 | ORF | 2075 | 5'-AATCTTTCACCACCCATAGT-3' |
| FKS1 | Promoter | −131 | 5'-TGTAGTTTGTGAGAAGGAGAAA-3' |
| FKS1 | Promoter | −7 | 5'-CCGTTGTATGAAAGACTTGATT-3' |
| FKS1 | ORF | 1939 | 5'-CCAATTAGAATTTTGTCCACCA-3' |
| FKS1 | ORF | 2047 | 5'-TAGCGATAACCAAACCTAAGAC-3' |
| QDR3 | Promoter | −111 | 5'-TAATAGCTGTGTCCTTGTATCC-3' |
| QDR3 | Promoter | 3 | 5'-CATGTTTATCGCTTTCTGACTT-3' |
| QDR3 | ORF | 1920 | 5'-CATGTTAAACGGTATGGGAAC-3' |
| QDR3 | ORF | 2044 | 5'-GTAAATCGTAGTTCTCTCTCCA-3' |
| SEC15 | Promoter | −75 | 5'-AATTAATACCTTTAACGAGCGT-3' |
| SEC15 | Promoter | 47 | 5'-ACCTGCTGAAAATCTTTTGAAA-3' |
| SEC15 | ORF | 1950 | 5'-GGAAATACGGTTATCCTCGATA-3' |
| SEC15 | ORF | 2072 | 5'-TGCCAGTCAATTTCAATAGTTT-3' |
| NAB6 | Promoter | −145 | 5'-CATCCAGAGAAGATATCCCAAA-3' |
| NAB6 | Promoter | −31 | 5'-GGATTCTTGCGAGTCTTGTT-3' |
| NAB6 | ORF | 1942 | 5'-TCAGACATAGGCAATAGAACAA-3' |
| NAB6 | ORF | 2051 | 5'-ATGTACTTAATGCTCTGAAGGA-3' |
| CHS5 | Promoter | −113 | 5'-CCCTTCAAGTTCTCCTTTCTAA-3' |
| CHS5 | Promoter | 11 | 5'-ACTGAAGACATTATTCGCTACT-3' |
| CHS5 | ORF | 1921 | 5'-GTTTTGTCCACTAAAGAAGCTA-3' |
| CHS5 | ORF | 2035 | 5'-CATTGAAGGCATCCATTAATCA-3' |
| KAR2 | Promoter | −80 | 5'-TCTAAAGATTAACGTGTTACTGT-3' |
| KAR2 | Promoter | 3 | 5'-CATGGTATGTTTGATACGCTTT-3' |
| KAR2 | ORF | 1930 | 5'-AAGGTCGCTTATCCAATTACTT-3' |
| KAR2 | ORF | 2027 | 5'-TAATCACCATCGTCATCTTCAT-3' |
| USA1 | Promoter | −83 | 5'-TGACGTACTTCAGATAAACACT-3' |
| USA1 | Promoter | 17 | 5'-GCTAGATATTCAGACATGTTGC-3' |
| USA1 | ORF | 1930 | 5'-CAAAGGCTATCGGTCTATTCTA-3' |
| USA1 | ORF | 2025 | 5'-CGATAGCACCTTGATAAATAGC-3' |
| SLA1 | Promoter | −112 | 5'-CAGAACGAATATTTAGCGCATA-3' |
| SLA1 | Promoter | 9 | 5'-CACAGTCATACTCTAGCTCTTT-3' |
| SLA1 | ORF | 1998 | 5'-TGATGTAAGCAATTGTCAAAGA-3' |
| SLA1 | ORF | 2085 | 5'-CATTGAGTTATTGATGTCAGGC-3' |
| SET1 | Promoter | −94 | 5'-CTGTTAGCAACCCTCAACTTA-3' |

*Continued on next page*

*Continued*

| Gene | Location | Position relative to ATG | Sequence |
|------|----------|--------------------------|----------|
| SET1 | Promoter | 9 | 5'-ATTTGACATTCTCTAAACGCAG-3' |
| SET1 | ORF | 1938 | 5'-ACATTTACTGAACGAAGAAACC-3' |
| SET1 | ORF | 2035 | 5'-TTTCGTCTTCTTCATCATGTTC-3' |
| HSF1 | Promoter | −91 | 5'-ATAAAGGCAAAGAGTTAGAGGT-3' |
| HSF1 | Promoter | 33 | 5'-ATTGGTCGTCCCTGTATTTG-3' |
| HSF1 | ORF | 1908 | 5'-TATAGACGAACAAGATGCAAGA-3' |
| HSF1 | ORF | 2021 | 5'-GAATTAGTGTTTGTCGAGGAAG-3' |

## Data analysis

We processed sequencing data essentially as described previously (*Moqtaderi et al., 2018*), mostly using Python 3 (www.python.org). After separating sequence reads from multiplexed libraries by barcode into output from individual samples, we removed adapter sequences from read ends and discarded reads with ambiguous bases and reads not starting with a T (corresponding to an A at the mRNA 3' end, potentially from polyadenylation). We counted and deleted consecutive Ts at the beginning of each read, saving the number of initial Ts for reference by appending it to the read ID. We mapped the first 17 nt of remaining sequence for each read to the *Saccharomyces cerevisiae* genome (version Sac cer3) using Bowtie [*Langmead et al., 2009*], allowing no mismatches and excluding non-unique matches. To ensure that we were working with post-transcriptionally adenylated RNA, we examined the genomic sequence immediately downstream of each mapped read, keeping only those reads for which the initial T count exceeded the number of consecutive As in the adjacent genomic sequence. Lastly, we scaled the remaining mapped reads for each replicate to a total of 25 million.

## End zone profiles, important parameters, and definitions

We assigned reads to a gene if they mapped within the 400 nt 3' UTR window downstream of its ORF. For each sample, we tabulated mRNA 3' isoform endpoint frequencies for all non-A positions within the first 400 nt of each 3'UTR. These isoform endpoint positions are numbered relative to the end of the associated ORF; for example, position 100 refers to the position 100 nt after the stop codon. We limited most of our analyses to the 2790 genes with ≥1000 normalized reads (combined from both biological replicates) in each of the 11 conditions/strains described in this work.

We constructed end zone profiles by setting the maximally expressed isoform (max isoform) in each gene's 3'UTR to 100 percent and linearly scaling expression levels of all other isoforms for that gene relative to this maximal value. The overall pattern of isoform expression over the 3'UTR constitutes a gene's end zone profile. 'Major isoforms' are any isoforms with expression levels equaling or exceeding 5% relative to the max isoform. The 'major end zone' comprises the region between the 5'-most and 3'-most major isoforms; the 'major end zone span' is its length. The 'weighted average isoform endpoint' for a gene is computed by adding up the endpoint positions (relative to the ORF end) of all reads mapping to its 3' UTR and dividing the result by the total number of reads summed.

## Percent coordinate usage analysis

For the 2790 genes analyzed in *Figure 1—figure supplement 1D*, *Figure 2—figure supplement 1*, *Figure 4—figure supplement 1A*, and *Figure 6—figure supplement 1*, we first calculated the total number of all non-A positions at each of the 400 locations (+1 to +400) of the 3' UTR. We then tabulated the total number of genes that had non-zero reads at each position within the 3' UTR. Finally, at each location in the 3'UTR, we divided the total number of genes with non-zero reads at that location by the total number of all non-A positions at the same coordinate and multiplied the resulting fraction by 100 to obtain the percentage of genes with reads at each coordinate.

## Correlations of biological replicates

We assessed the reproducibility of biological replicates in several ways. First, we compared the total expression by gene between replicates. For each gene, we obtained the total expression level by summing all reads mapping anywhere within the first 400 nt after the ORF. For each of our 11 experimental conditions/strains, we computed the Pearson correlation of total gene expression at a minimum of 5000 genes across two biological replicates (*Figure 1—figure supplement 1A*, panel 1). Second, we compared the expression of 25,000–75,000 individual 3' isoforms genome-wide across replicates of the same 11 conditions/strains (*Figure 1—figure supplement 1A*, panel 2), omitting isoforms with fewer than 10 reads. Third, we compared end zone profiles of individual genes between biological replicates. For this, we analyzed the 2790 genes whose combined expression in both biological replicates was ≥1000 normalized reads in all 11 conditions/strains. For each gene, we computed the Pearson R coefficient across biological replicates by correlating scaled read counts by position for the entire 400 nt 3'UTR (*Figure 1—figure supplement 1*, panel 3). Combined 3' READS data from both biological replicates of exponentially growing JGY2000 compare favorably to our previously published no-DMS control dataset for DREADS, a closely-related assay that captures structural information on individual mRNA 3'UTR isoforms (*Moqtaderi et al., 2018*; data not shown).

## Classification of genes by sensitivity to pol II elongation rate perturbations

For each of the slow or fast Pol II strains (JZY5, JZY6, JZY14, and JZY15), we constructed an error model to identify and classify genes whose end zone profiles are significantly shifted (either upstream or downstream) due to changes in the Pol II elongation rate. First, we computed individual percentile coordinates (10%, 25%, 50%, 75% and 90%) for all 2790 genes in each biological replicate for all the Pol II mutant strains and the exponentially growing JGY2000. These coordinates represent 3'UTR locations at which the indicated percentage of total reads occurs upstream of (and including) the calculated coordinate.

For every gene in each strain, we then subtracted the individual percentile coordinates in each biological replicate from each other to obtain raw error values at all five percentile coordinates. Individual raw error values from the four Pol II elongation rate strains were then separately averaged with the corresponding raw error values from the exponentially growing JGY2000 dataset, and the rounded absolute values of those measurements were termed either the 10th-, 25th-, 50th-, 75th-, or 90th-percentile errors. The frequency distributions of the 10th-, 25th-, 50th-, 75th-, and 90th-percentile errors for JZY4, JZY5, JZY14, and JZY15 (2790 values/percentile for each strain) were tabulated after dividing all non-zero errors by two in order to account for the fact that the error could be either positive or negative.

Using the 10th percentile parameter as an example, cumulative probabilities at each error value $x$ were calculated by the following equation,

$$P(x) = \frac{\sum\limits_{i=x}^{\text{Max}(x)} f(i)}{2,790}$$

where Max($x$) represents the maximum observed 10th-percentile error value and $f(i)$ is the frequency of the error $i$ in the distribution. Therefore, the probability that an experimentally observed net shift of magnitude $|k|$ in the 10th percentile coordinate is due to pure chance is given by $P(|k|)$. In cases where $|k| > $ Max($x$), P($|k|$) was assigned the lowest non-zero probability of $3.58 \times 10^{-4}$, which equals to 1/2,790. Cumulative probabilities were calculated for the remaining (25th, 50th, 75th, and 90th) percentile categories as described above.

Experimental net shift values ($k$) were calculated as follows. Using JZY5 as an example (the same methodology applies to JZY6, JZY14, and JZY15), we (1) subtracted the individual percentile coordinates (see above) at every gene in each biological replicate of JZY5 from their corresponding values in the biological replicates of exponentially growing JGY2000, (2) divided the values by two and (3) rounded the difference to the nearest whole number to obtain the raw shift values. From raw shift values we then subtracted the corresponding raw error values (see above) to obtain net shift values ($k$). $k$ was set to zero for all cases where the raw error value was greater than the corresponding raw shift value.

The five net shift values (per gene) represent error-subtracted measures of poly(A) position shifts at the indicated percentiles. Negative $k$ values represent upstream shifts in poly(A) usage of JZY5 relative to exponentially-growing JGY2000. Conversely, positive $k$ values reflect greater downstream poly(A) utilization at the indicated percentile categories in JZY5 versus the strain with the normal Pol II elongation rate (exponentially-growing JGY2000).

For each $|k|$, a probability $P(k)$ was computed based on the error model (see above). A single-gene probability value $P(g)$ was computed by multiplying all the individual $P(k)$ probabilities at the five percentile categories by one another and then by five in order to correct for multiple hypotheses (*Dunn, 1961*).

For each gene, we calculated two additional parameters: a cumulative net shift and the net number of positions shifted. The former parameter represents the sum of all net shifts ($\Sigma k$) at the five percentile coordinates. A positive cumulative net shift represents an overall downstream shift in JZY5 poly(A) sites, while a negative number implies a greater prevalence of shorter 3'UTR isoforms in JZY5 relative to exponentially growing JGY2000. The net number of positions shifted is a measure of the total number (as well as the direction of the shift; see below) of the five percentile coordinates that had a non-zero net shift. It was computed by assigning each of the five gene-specific percentile coordinates a value of either −1 (representing a net upstream shift, or negative $k$ value, at that position), +1 (implying a net downstream shift, or a positive $k$ value, at that position), or 0 (no net shift). The sum of the five values for each gene is the net number of shifted positions. A negative net number of positions shifted means that the overall shift in JZY5 poly(A) sites is more likely to be upstream, while a positive net number of positions shifted indicates a distal poly(A) shift in JZY5 relative to exponentially growing JGY2000. This parameter, along with the cumulative overall shift, is especially helpful in assigning genes to specific categories in more complex cases where some of the $k$ values point to shifts in opposite directions.

## Initial classification of poly(A) shift direction by gene

For each of the four Pol II strains (JZY5-6, JZY14-15), we first assigned the 2790 genes into one of three categories based on the comparison of each gene's end zone profile in the Pol II mutant strain to its profile in the exponentially growing JGY2000. The three categories consisted of (1) genes whose poly(A) sites were shifted upstream in a given Pol II mutant strain relative to exponentially growing JGY2000 ('upshifted'), (2) genes whose end zone profiles in the given mutant Pol II strain were shifted downstream relative to exponentially growing JGY2000 ('downshifted'), and (3) genes whose end zone profiles were not classified as either upstream or downstream ('other'). Genes were categorized as upshifted if they met the all of following criteria: (a) a negative cumulative net shift, (b) a negative net number of positions shifted and (c) a $P(g)$ value <0.01. In order to be categorized as downshifted, a gene had to possess (a) a positive cumulative net shift, (b) a positive net number of positions shifted and (c) a $P(g)$ value <0.01.

## Combined classification of poly(A) shift behavior by gene across multiple strains

We classified each of the 2790 genes by combining data from the individual, strain-specific categorization (see above) into one of four groups: 'Upstream', 'Downstream', 'Neutral' and 'Other'. In the combined classification, a gene was classified as Upstream if it was upshifted in both Pol II elongation rate-defective strains (JZY5 and JZY6; see above), irrespective of its behavior in the two fast Pol II strains (JZY14 and JZY15). Similarly, a gene was classified as Downstream if it was downshifted in the each of the two Pol II strains with the fast elongation rate, without regard for its behavior in the slow Pol II strains. A gene was called Neutral if it was classified as 'other' in all four of the Pol II elongation rate mutant strains. All genes that didn't fit any of the criteria above (304 out of 2,790; for example, genes which were upshifted or downshifted in only one strain, etc.) were classified as Other. Finally, we noticed that a large proportion of Downstream genes were also classified as Upstream (*Figure 4*), and we named this sub-category (which represents the intersection of the Upstream and Downstream groups) 'Both'.

## Conservation of endpoints

Calculations of probabilities that major isoform positions in JZY5, JZY6, JZY14, JZY15 and diauxic JGY2000 overlap with those in exponentially grown JGY2000 were conducted in identical, pairwise fashion. First, for each of the 2790 genes, we identified the portion of the 3' UTR in which meaningful polyadenylation was observed in any of our 11 strains/conditions. This 'combined major end zone' is the union of the gene's major end zones in every condition tested. Its 5' boundary is the most ORF-proximal of all major isoforms observed in any of the 11 conditions. Similarly, the 3' boundary is the most ORF-distal of all major isoforms found in the 11 conditions/strains.

For each gene (using JZY5 as an example), the cumulative probability $P(q)$ for major isoform overlap between JZY5 and exponentially growing JGY2000 is given by the hypergeometric distribution.

$$\text{IF } \alpha \geq \beta: \quad P(q) = \sum_{i=c}^{\beta} \frac{\binom{\beta}{i}\binom{N-\beta}{\alpha-i}}{\binom{N}{\alpha}} \quad IF \beta \geq \alpha: \quad P(q) = \sum_{i=c}^{\alpha} \frac{\binom{\alpha}{i}\binom{N-\alpha}{\beta-i}}{\binom{N}{\beta}}$$

where N is the number of non-A positions in the combined major end zone, $\alpha$ is the number of major isoforms in exponentially growing JGY2000, $\beta$ is the number of major isoforms in JZY5, and $c$ is the number of major isoform positions in common between the two strains. In calculations where the entire 3'UTR is assumed to be permissive for polyadenylation (i.e. major poly(A) isoforms are not limited to the combined major end zone window; *Figure 3—figure supplement 1A*), N is replaced by the number of non-A positions within each gene's 400-nt 3'UTR. Probability calculations for all other strains listed above were performed exactly as described for JZY5.

## Nucleotide frequency composition analysis

Nucleotide frequencies in exponentially-growing JGY2000, diauxic JGY2000, JZY5, and JZY14 were tabulated for max isoform positions. The $-1$ position refers to the last genomically-encoded nucleotide (i.e., the base immediately upstream of the cleavage/polyadenylation site) within each isoform. Therefore, all positive positions (i.e. positions to the right of the cleavage/polyadenylation site) are not encoded in the actual isoforms. Nucleotide frequencies were computed by summing up the number of A's, C's, G's and U's at each position within a category, dividing these numbers by the total number of genes within the category and multiplying the resulting fraction by 100. The 'overall' category consisted of 2790 genes, while the Upstream and Downstream categories contained 1898 and 605 genes, respectively. The 'Both' sub-category, consisting of genes whose end zones are shifted upstream in both slow Pol II mutant strains and shifted downstream in both fast Pol II strains, contained a total of 462 genes. Finally, the 'Neutral' category comprised 445 genes.

## Acknowledgements

We thank Catherine Maddox for excellent technical assistance and Craig Kaplan for helpful advice on constructing the Pol II mutant strains. This work was supported by grants to KS from the National Institutes of Health (GM30186 and GM131801).

## Additional information

### Competing interests

Kevin Struhl: Senior editor, *eLife*. The other authors declare that no competing interests exist.

### Funding

| Funder | Grant reference number | Author |
| --- | --- | --- |
| National Institutes of Health | GM 30186 | Joseph V Geisberg<br>Zarmik Moqtaderi<br>Kevin Struhl |
| National Institutes of Health | GM 131801 | Joseph V Geisberg<br>Zarmik Moqtaderi |

Kevin Struhl

The funders had no role in study design, data collection and interpretation, or the decision to submit the work for publication.

## Author contributions
Joseph V Geisberg, Zarmik Moqtaderi, Conceptualization, Data curation, Software, Formal analysis, Validation, Investigation, Visualization, Methodology, Writing - original draft, Writing - review and editing; Kevin Struhl, Conceptualization, Formal analysis, Supervision, Funding acquisition, Writing - original draft, Project administration, Writing - review and editing

## Author ORCIDs
Zarmik Moqtaderi ⓘD https://orcid.org/0000-0002-2785-7034
Kevin Struhl ⓘD https://orcid.org/0000-0002-4181-7856

## Decision letter and Author response
Decision letter https://doi.org/10.7554/eLife.59810.sa1
Author response https://doi.org/10.7554/eLife.59810.sa2

# Additional files
## Supplementary files
• Transparent reporting form

## Data availability
Sequencing data has been deposited in GEO under accession code GSE151196.

The following dataset was generated:

| Author(s) | Year | Dataset title | Dataset URL | Database and Identifier |
|---|---|---|---|---|
| Geisberg JV, Moqtaderi Z, Struhl K | 2020 | The transcriptional elongation rate regulates alternative polyadenylation in yeast | http://www.ncbi.nlm.nih.gov/geo/query/acc.cgi?acc=GSE151196 | NCBI Gene Expression Omnibus, GSE151196 |

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
