## [Decision Letter]

**Acceptance summary:**

The authors have explored how a metabolic state change in budding yeast, known as the diauxic shift, impacts global cleavage and polyadenylation. Overall, this work reveals that environmental cues can broadly impact alternative polyadenylation that is likely manifested through alterations in Pol II elongation rates. This work is important because it reveals a previously unsuspected interplay between Pol II function and RNA processing.

**Decision letter after peer review:**

Thank you for submitting your article "The transcriptional elongation rate regulates alternative polyadenylation in yeast" for consideration by *eLife*. Your article has been reviewed by James Manley as the Senior Editor, a Reviewing Editor, and three reviewers. The following individuals involved in review of your submission have agreed to reveal their identity: Nick J Proudfoot (Reviewer #1).

The reviewers have discussed the reviews with one another and the Reviewing Editor has drafted this decision to help you prepare a revised submission.

Summary:

The authors have explored how a metabolic state change in budding yeast, known as the diauxic shift, impacts global cleavage and polyadenylation. In this research article, they determine that upon diauxic shift there is a broad change in poly(A) site usage to favor cleavage and polyadenylation sites more proximal to stop codons. The upstream shift of poly(A) site usage in the diauxic condition, interestingly, matches well with that observed in yeast expressing Pol II slow mutants. Because Hpr1 and Spt4 mutants, which have processivity issues, do not display such an upstream shift, the authors conclude that it is elongation rate, rather than processivity, that leads to the upstream shift in diauxic condition. Overall, this work reveals that environmental cues can broadly impact alternative polyadenylation that is likely manifested through alterations in Pol II elongation rates underscoring the interplay between Pol II function and RNA processing.

Below is a relatively short list of essential revisions that are believed to be addressable in a contained amount of time. The most prominent of those concerns is the need to provide stronger data indicating that Pol II elongation rate is indeed reduced during diauxic shift – a point central to the overall manuscript conclusion.

Essential revisions:

Experimental:

1) The Pol II fast mutants do not seem to lend much support to their model. Is this because of mitigation of alternative polyadenylation profile changes by RNA degradation? For example, if long isoforms are generated but are also preferentially degraded, the downstream shift may not be obvious. Also, the authors need to examine this possibility. The authors need to check Pol II fast mutants in diauxic condition to see if the upstream shift could be inhibited. This would greatly strengthen their conclusion. (This point is assuming that research has resumed.)

2) Figure 6 is a clear-cut data set showing that diauxic shift does indeed correlate with TSS proximal enhanced Pol II density over genes and that this is unaffected by loss of factors previously ascribed to processivity effects, rather than simply Pol II speed. However, some simple Pol II ChIPs here as well as bioinformatic Pol II occupancy measures would be helpful (Figure 6A and B).

Textual:

1) Figure 1 shows that diauxic shift as compared to simply different growth medium, switches many Pol II transcribed genes into usage of 5' located PAS. However, do these PAS switches affect mRNA levels of affected genes, i.e. does their gene expression change. Also does this correlate with gene ontology for changing the metabolic needs of the cell?

2) The bioinformatic data shown in Figure 3 that preferred PAS sites are non-random (presumably sequence specific) is somewhat over the top and that really these bioinformatic data are of less interest (maybe good for supplementary data?).

3) Improve the readability of Figure 5 as multiple reviewers found it hard to follow. A simple table of base composition would suffice rather than the hard-to-see graphical data across PAS sequences. Ideally the significance of a reduction in U (and C) richness correlating with PAS speed sensitivity needs to be tested by direct mutation? Also, they need to compare the same poly(A) sites that are upregulated or downregulated in diauxic condition vs. other conditions.

[Editors' note: further revisions were suggested prior to acceptance, as described below.]

Thank you for re-submitting your article "The transcriptional elongation rate regulates alternative polyadenylation in yeast" for consideration by *eLife*. Your revision has been re-reviewed by James Manley as the Senior Editor, a Reviewing Editor, and two reviewers. The reviewers have opted to remain anonymous.

There was a mixed response by the reviewers upon inspection of your revision. Ultimately, through discussion, a consensus was reached to request a revision that addresses their concerns. We stress that this revision need only be textual and agree with the reviewers that some aspects of the manuscript need additional clarification. Provided reasonable modifications to the text are made, we anticipate not sending this revision back to them. Rather than summarize their reviews, which are short, they are appended below.

Reviewer #2:

Geisberg et al. provided a forceful response to the previous review. Overall, while I am not fully convinced by the arguments presented, I am not in favor of prolonging the review, especially in the current difficult research situation caused by the pandemic. After all, the authors have done a solid work to generate substantial data that are useful to the community. I therefore suggest that the authors incorporate some of their response in the paper so that readers would not miss the true significance and novelty of this work. Specifically, they may want to emphasize (1) the true novelty of this work is the second theme, and (2) the features of speed-sensitive 3'UTRs. Regarding other potential explanations for poly(A) site usage in diauxic conditions, they may want to mention that alternation of some key 3' end processing factors may lead to similar changes.

Reviewer #3:

This is a revised manuscript. The editor and reviewers previously concluded that the authors needed "to provide stronger data indicating that Pol II elongation rate is indeed reduced during diauxic shift" to support the authors' conclusions, which the authors have not done. Nor have they adjusted the text of the manuscript to reflect this gap. Instead the authors argue in the rebuttal that the reviewers and editor confused two separate themes with theme one being that slow polymerase causes an upstream shift in poly(A) sites and theme 2 being that poly(A) sites shift upstream during diauxic shift. However, the manuscript continues to prominently suggest a mechanistic link between the two. The abstract literally ends with the bottom line that "Pol II elongation speed is important.…. for regulating poly(A) patterns in response to environmental conditions." This is not the only prominent occurrence where the authors draw very strong causal relationship (e.g. in subsection “Yeast cells containing Pol II derivatives with slow elongation rates show a poly(A) pattern that strikingly resembles the pattern in diauxic shift” where the authors say "strongly suggests a mechanistic relationship"; in subsections "Evidence that pol II elongation rate is decreased in diauxic conditions", “Pol II elongation rate, not processivity, is important for polyadenylation patterns” where the authors say "is the cause"; and in subsection “Evidence that regulated polyadenylation during the diauxic shift is due to decreased elongation rate” where the authors say "due to"). While a manuscript presenting two separate, but related, themes would be acceptable, that is not what the current manuscript does.

Most other comments have been successfully addressed.

---

## [Author Response]

Summary:The authors have explored how a metabolic state change in budding yeast, known as the diauxic shift, impacts global cleavage and polyadenylation. In this research article, they determine that upon diauxic shift there is a broad change in poly(A) site usage to favor cleavage and polyadenylation sites more proximal to stop codons. The upstream shift of poly(A) site usage in the diauxic condition, interestingly, matches well with that observed in yeast expressing Pol II slow mutants. Because Hpr1 and Spt4 mutants, which have processivity issues, do not display such an upstream shift, the authors conclude that it is elongation rate, rather than processivity, that leads to the upstream shift in diauxic condition. Overall, this work reveals that environmental cues can broadly impact alternative polyadenylation that is likely manifested through alterations in Pol II elongation rates underscoring the interplay between Pol II function and RNA processing.Below is a relatively short list of essential revisions that are believed to be addressable in a contained amount of time. The most prominent of those concerns is the need to provide stronger data indicating that Pol II elongation rate is indeed reduced during diauxic shift – a point central to the overall manuscript conclusion.

1) The paper has two major themes, but the reviewers have focused on the first theme, namely regulated polyadenylation in response to an environmental condition via Pol II speed. They largely overlooked the second theme, namely a detailed analysis of how Pol II speed affects polyadenylation. The few papers that investigate this second theme are much less advanced, typically using only a single slow Pol II derivative, not done at the nucleotide level (i.e. mRNA isoforms), and providing no information on the difference between speed-sensitive vs. speed-insensitive 3’ UTRs. As such, some of the results that the reviewers considered to be peripheral (which they are for theme 1) are critical for theme 2 and represent a significant advance over current knowledge. We previously considered dividing the paper into 2 back-to-back short papers, one for each theme. However, the two themes are both important and clearly related, which is why we presented them together.

2) Regarding the request to “provide stronger data indicating that Pol II elongation rate is indeed reduced during the diauxic shift,” we believe that our results and conclusions are compelling, even if below the level of formal proof (which we do not claim and which rarely happens in any paper). The poly(A) patterns under diauxic conditions and in two slow Pol II mutants are remarkably similar over thousands of genes. Diauxic conditions clearly show decreased Pol II processivity, a known feature of slow Pol II. Processivity per se is ruled out by the *spt4* and *hpr1* deletion experiments. I can’t think of anything plausible that could explain all these data other than our conclusion that Pol II elongation rate is reduced in diauxic conditions. If the reviewers can come up with a plausible alternative, we would be happy to mention it.

Furthermore, there are technical and conceptual problems in trying to “provide stronger data.” Technically, I don’t know how we can measure Pol II speed in diauxic conditions. We can’t use our glucose-shutoff method. Conceptually, even if we could measure speed, it would be impossible to disentangle a true effect on Pol II speed vs. a general slowdown of molecular processes in diauxic conditions, in which cells are barely growing. Trying to correct/normalize the effects of cell growth on molecular processes is a quagmire many have encountered and few have solved. The processivity experiments in the paper get around this, because it is the pattern that is affected, not the absolute rate.

Essential revisions:Experimental:1) The Pol II fast mutants do not seem to lend much support to their model. Is this because of mitigation of alternative polyadenylation profile changes by RNA degradation? For example, if long isoforms are generated but are also preferentially degraded, the downstream shift may not be obvious. Also, the authors need to examine this possibility. The authors need to check Pol II fast mutants in diauxic condition to see if the upstream shift could be inhibited. This would greatly strengthen their conclusion. (This point is assuming that research has resumed.)

The reviewers are correct that the fast Pol II mutants are largely irrelevant to the regulation in response to the diauxic shift (i.e. theme 1), and we did not use them at all for this purpose. As such, the other aspects of this comment are also irrelevant for theme 1. However, they are critical to theme 2, which was largely overlooked by the reviewers.

The suggestion that long isoforms in the fast Pol II mutants might be preferentially degraded is extremely unlikely based on what we and others have already published about mRNA decay in yeast (Geisberg et al., 2014; Gupta et al., 2014). Our previous work identified many hundreds of stabilizing and destabilizing elements within 3’UTRs. These elements can occur anywhere within a given 3’UTR, meaning that longer isoforms arising from a given gene can be either more or less stable than shorter isoforms. Furthermore, a completely independent series of experiments clearly demonstrated there was no correlation between 3’UTR length and stability (Gupta et al., 2014). While longer mRNA isoform destabilization had been reported (and varies in its extent) in cancer cells and other cell lines (Mayr and Bartel, 2009; Lin et al., 2012; Spies et al., 2013), the results from both studies described above are completely inconsistent with a general destabilization of longer 3’ isoforms. Furthermore, the key point of Figure 3 is that the same poly(A) sites are used in all Pol II derivatives, so mRNA stability of these isoforms will be the same in all strains. Nevertheless, in response to this comment, the revised paper now discusses issues related to mRNA stability.

The suggested experiment to check fast Pol II mutants under diauxic conditions would yield uninterpretable results. Pitting an upstream-shifting condition against a downstream-shifting condition merely asks which is more potent and provides no mechanistic information. Typically, such experiments give intermediate effects, which in the case here would be an upstream shift that is somewhat less pronounced than a wt strain under diauxic conditions (especially because the upstream shift in diauxic conditions is much stronger than the downstream shift caused by the fast Pol II mutant). However, the precise answer could be anything, and hence it would provide no information to strengthen any conclusion in the paper.

2) Figure 6 is a clear-cut data set showing that diauxic shift does indeed correlate with TSS proximal enhanced Pol II density over genes and that this is unaffected by loss of factors previously ascribed to processivity effects, rather than simply Pol II speed. However, some simple Pol II ChIPs here as well as bioinformatic Pol II occupancy measures would be helpful (Figure 6A and B).

We don’t understand this comment about doing Pol II ChIP experiments. Figure 6A is a Pol II ChIP experiment, with the results plotted with respect to the upstream shift in Figure 6B. Processivity and speed experiments for the Spt4 and Hpr1 mutants were published 15 years ago (Mason and Struhl, 2005), and they involved Pol II ChIP.

Textual:1) Figure 1 shows that diauxic shift as compared to simply different growth medium, switches many Pol II transcribed genes into usage of 5' located PAS. However, do these PAS switches affect mRNA levels of affected genes, i.e. does their gene expression change. Also does this correlate with gene ontology for changing the metabolic needs of the cell?

Because the paper is concerned with poly(A) profiles, the results are normalized for each gene with the maximal isoform given a value of 100. The reviewers are correct that this presentation ignores the changes in expression level under the various conditions. As such, we now add a paragraph about this (the expression values are present in the Excel files in our GEO submission). As expected, whether or not a gene is regulated under a given conditions does not affect its poly(A) profile. Under all conditions except diauxic, the poly(A) profiles are the same even though expression levels of many genes are regulated. Under diauxic conditions, the vast majority of genes shift in the same manner, even though only a subset is regulated at the expression level.

2) The bioinformatic data shown in Figure 3 that preferred PAS sites are non-random (presumably sequence specific) is somewhat over the top and that really these bioinformatic data are of less interest (maybe good for supplementary data?).

Figure 3 makes the important point that upstream and downstream shifts involve the same poly(A) sites, just a rebalancing of how much they are used. While this might (or might not) have been expected, it is important to demonstrate, especially as it has mechanistic implications (e.g. see above response about suggested experiments). Moreover, it is a useful piece of information for theme 2, and in this regard, we are unaware of anyone looking at this issue at the nucleotide level on a transcriptome scale. As such, we believe that it merits a proper figure.

3) Improve the readability of Figure 5 as multiple reviewers found it hard to follow. A simple table of base composition would suffice rather than the hard-to-see graphical data across PAS sequences. Ideally the significance of a reduction in U (and C) richness correlating with PAS speed sensitivity needs to be tested by direct mutation? Also, they need to compare the same poly(A) sites that are upregulated or downregulated in diauxic condition vs. other conditions.

As requested, we simplified Figure 5 to show how speed-sensitive 3’UTRs differ in base composition from neutral 3’UTRs. Specifically, we showed this only for the wild-type strain, meaning only 2 lines for each nucleotide. This result is identical for all other comparisons including slow only and fast only Pol II, as well as the diauxic condition, and the data for all these other comparisons is now part of the supplemental figure associated with Figure 5. We think it important to keep the form of the original figure because the level of nucleotide preferences varies over the range examined, and the readers should see this instead of a Table that is a simple summary of preferences at an arbitrary location(s).

[Editors' note: further revisions were suggested prior to acceptance, as described below.]

(…)Reviewer #2:Geisberg et al. provided a forceful response to the previous review. Overall, while I am not fully convinced by the arguments presented, I am not in favor of prolonging the review, especially in the current difficult research situation caused by the pandemic. After all, the authors have done a solid work to generate substantial data that are useful to the community. I therefore suggest that the authors incorporate some of their response in the paper so that readers would not miss the true significance and novelty of this work. Specifically, they may want to emphasize on (1) the true novelty of this work is the second theme, and (2) the features of speed-sensitive 3'UTRs. Regarding other potential explanations for poly(A) site usage in diauxic conditions, they may want to mention that alternation of some key 3' end processing factors may lead to similar changes.

1) As requested, we now consider the suggestion that alteration of some 3’ processing factor might account for the poly(A) shift in diauxic conditions. We thank the reviewer for bringing up an alternative explanation to our conclusion, which is very helpful for readers of the paper. However, as discussed in a new paragraph, this suggested model is extremely unlikely for two reasons. First, this suggested model would require such a altered 3’ processing factor to cause a virtually identical poly(A) pattern to those of two different slow Pol II mutants over thousands of genes. Second, if the shift in diauxic conditions is due to a 3’ processing factor, why is there such a pronounced Pol II processivity defect, a hallmark of reduced Pol II speed? In addition and unbeknownst to the reviewers, we actually have looked at poly(A) patterns in numerous 3’ processing and transcriptional elongation factors, and we have never seen an upstream shift like those in diauxic and slow Pol II conditions. These data are unpublished and not fully analyzed, but this result is clear.

2) As requested, we have added text to the subsection “Pol II elongation rate, not Pol II processivity, affects poly(A) site selection” to further emphasize theme 2.

Reviewer #3:This is a revised manuscript. The editor and reviewers previously concluded that the authors needed "to provide stronger data indicating that Pol II elongation rate is indeed reduced during diauxic shift" to support the authors' conclusions, which the authors have not done. Nor have they adjusted the text of the manuscript to reflect this gap. Instead the authors argue in the rebuttal that the reviewers and editor confused two separate themes with theme one being that slow polymerase causes an upstream shift in poly(A) sites and theme 2 being that poly(A) sites shift upstream during diauxic shift. However, the manuscript continues to prominently suggest a mechanistic link between the two. The abstract literally ends with the bottom line that "Pol II elongation speed is important.…. for regulating poly(A) patterns in response to environmental conditions." This is not the only prominent occurrence where the authors draw very strong causal relationship (e.g. in subsection “Yeast cells containing Pol II derivatives with slow elongation rates show a poly(A) pattern that strikingly resembles the pattern in diauxic shift” where the authors say "strongly suggests a mechanistic relationship"; in subsections "Evidence that pol II elongation rate is decreased in diauxic conditions", “Pol II elongation rate, not processivity, is important for polyadenylation patterns” where the authors say "is the cause"; and in subsection “Evidence that regulated polyadenylation during the diauxic shift is due to decreased elongation rate” where the authors say "due to"). While a manuscript presenting two separate, but related, themes would be acceptable, that is not what the current manuscript does.

As requested, we have softened the statements in a few places and also explicitly discussed an alternative explanation of the data (see point 1 of reviewer 2). However, we find it difficult to respond to this re-review. While there was a general request “to provide stronger data indicating that Pol II elongation rate is indeed induced during diauxic shift,” no specific experiments were suggested in the “essential revisions” of the first review and none are suggested in the re-review. Furthermore, in our response to the original reviews, we provided detailed arguments about (1) why our conclusion is compelling (even if not formally proven, which we never claim), (2) why a direct measurement of Pol II speed in diauxic conditions is technically impossible, and (3) why any such measurement could not be interpreted due to complications related to slow growth. Reviewer 3 did not challenge or address these arguments.

I think we accurately stated the strength of our conclusions and have never described them with terms such as “proven”, “conclusive” or “definitive”. “Evidence that” is clearly a qualified statement. The examples of “is the cause” and “due to” are misleading because in all of these examples were qualified by strongly suggest (which we have now softened to suggest) in the same sentence.

There appears to be some confusion about what themes 1 and 2 are. Theme 1 is about regulated polyadenylation and includes upstream shifts in *both* diauxic and slow Pol II mutants and the processivity experiments. Theme 2 concerns mechanistic information about the relationship of Pol II speed to polyadenylation and includes the fast Pol II data, gene specificity and sequence determinants of speed dependence, and rebalancing of sites upon the shift and the absence of new sites. We mentioned the two themes in our initial response because theme 2 was largely overlooked in the original reviews even though we (and reviewer 2) thought it an important aspect of the paper. It had nothing to do with regulated polyadenylation (theme 1).